

# Future changes in isoprene-epoxydiol-derived secondary organic aerosol (IEPOX-SOA) under the shared socioeconomic pathways: the importance of explicit chemistry

Duseong S. Jo[1,2,3], Alma Hodzic[3], Louisa K. Emmons[3], Simone Tilmes[3], Rebecca H. Schwantes[3,#,△],
Michael J. Mills[3], Pedro Campuzano-Jost[1,2], Weiwei Hu[1,2,*], Rahul A. Zaveri[4], Richard C. Easter[4],
Balwinder Singh[4], Zheng Lu[5], Christiane Schulz[6,7], Johannes Schneider[6], John E. Shilling[4], Armin
Wisthaler[8,9], and Jose L. Jimenez[1,2]

[1] Cooperative Institute for Research in Environmental Sciences (CIRES), University of Colorado, Boulder, CO, USA
[2] Department of Chemistry, University of Colorado, Boulder, CO, USA
[3] National Center for Atmospheric Research, Boulder, CO, USA
[4] Atmospheric Sciences and Global Change Division, Pacific Northwest National Laboratory, Richland, WA, USA
[5] Department of Atmospheric Sciences, Texas A&M University, College Station, Texas
[6] Particle Chemistry Department, Max Planck Institute for Chemistry, Mainz, Germany
[7] Leibniz Institute for Tropospheric Research, Leipzig, Germany
[8] Department of Chemistry, University of Oslo, Oslo, Norway
[9] Institute for Ion Physics and Applied Physics, University of Innsbruck, Innsbruck, Austria
#Now at: Cooperative Institute for Research in Environmental Sciences, University of Colorado, Boulder, CO, USA
△Now at: Chemical Sciences Laboratory, National Oceanic and Atmospheric Administration, Boulder, CO, USA
*Now at: State Key Laboratory of Organic Geochemistry, Guangzhou Institute of Geochemistry, Chinese Academy of
Sciences, Guangzhou, China

*Correspondence to*: Duseong S. Jo (duseong.jo@colorado.edu)

**Abstract.** Secondary organic aerosol (SOA) is a dominant contributor of fine particulate matter in the atmosphere, but the complexity of SOA formation chemistry hinders the accurate representation of SOA in models. Volatility-based SOA parameterizations have been adopted in many recent chemistry modeling studies and have shown a reasonable performance compared to observations. However, assumptions made in these empirical parameterizations can lead to substantial errors when applied to
future climatic conditions as they do not include the mechanistic understanding of processes but are rather fitted to laboratory studies of SOA formation. This is particularly the case for SOA derived from isoprene epoxydiols (IEPOX-SOA), for which we have a higher level of understanding of the fundamental processes than is currently parameterized in most models. We predict future SOA concentrations using an explicit mechanism, and compare the predictions with the empirical



parameterization based on the volatility basis set (VBS) approach. We then use the Community Earth System Model 2 (CESM2.1.0) with detailed isoprene chemistry and reactive uptake processes for the middle and end of the 21st century under four Shared Socioeconomic Pathways (SSPs): SSP1-2.6, SSP2-4.5, SSP3-7.0, and SSP5-8.5. With the explicit chemical mechanism, we find that IEPOX-SOA is predicted to increase on average under all future SSP scenarios, however with some variability in the

results depending on regions and the scenario chosen. Isoprene emission is the main driver of IEPOX-SOA changes in the future climate, but IEPOX-SOA yield from isoprene emission also changes by up to 50% depending on the SSP scenario, in particular due to different sulfur emissions. We conduct sensitivity simulations with and without $CO_2$ inhibition of isoprene emissions that is highly uncertain, which results in a factor of two differences in the predicted IEPOX-SOA global burden, especially for

the high-$CO_2$ scenarios (SSP3-7.0 and SSP5-8.5). Aerosol pH also plays a critical role in the IEPOX-SOA formation rate, requiring accurate calculation of aerosol pH in chemistry models. On the other hand, isoprene SOA calculated with the VBS scheme predicts nearly constant SOA yield from isoprene emission across all SSP scenarios, as a result, it mostly follows isoprene emissions regardless of region and scenario. This is because the VBS scheme does not consider heterogeneous chemistry, in

other words, there is no dependency on aerosol properties. The discrepancy between the explicit mechanism and VBS parameterization in this study is likely to occur for other SOA components as well, which may also have dependencies that cannot be captured by VBS parameterizations. This study highlights the need for more explicit chemistry, or for parameterizations that capture the dependence on key physico-chemical drivers when predicting SOA concentrations for climate studies.



# 1 Introduction

Secondary organic aerosol (SOA) contributes substantial mass fractions of submicron particle concentrations globally (Zhang et al., 2007), but it is difficult to predict due to its complex sources and behavior in the atmosphere (Tsigaridis et al., 2014). There are typically many thousands of organic species and reactions involved depending on the precursor hydrocarbon (Lannuque et al., 2018), which necessitates the use of parameterizations in chemistry-climate SOA simulations. Over the last two decades, there has been much progress in understanding chemistry of SOA (Ziemann and Atkinson, 2012; Glasius and Goldstein, 2016; Shrivastava et al., 2017; Bianchi et al., 2019), which has helped chemical transport models to develop more advanced SOA parameterizations (Hodzic et al., 2016; Tsimpidi et al., 2018; Schervish and Donahue, 2019).

Parameterizations range from simplified empirical methods (Hodzic and Jimenez, 2011) to volatility basis set (VBS) approaches of various complexity (Donahue et al., 2006, 2012) which can calculate not only mass concentrations but also oxidation states (Tsimpidi et al., 2018). A recent evaluation of global OA schemes reported that a simple parameterization showed similar performance compared to explicit chemistry and a complex parameterization scheme, in terms of bias and variability (Pai et al., 2019). This can be explained by the fact that simple parameterizations are based on observational constraints from field campaigns (Hodzic and Jimenez, 2011; Kim et al., 2015). However, even though two different SOA schemes simulated similar SOA concentrations in present-day atmospheric conditions, they can have different responses to not only preindustrial conditions (Tilmes et al., 2019) but also future emissions and climate changes and provide different performance in predicting future SOA concentrations, as discussed below.

In order to obtain realistic predictions of future SOA changes, all the variables affecting SOA concentrations should be considered. However, parameterizations inevitably omit some important processes, for example, the two-product or VBS approaches conventionally used in chemical transport models do not consider acid-catalyzed multiphase chemistry (Marais et al., 2016; Zhang et al., 2018a)





and autoxidation (Bianchi et al., 2019; Schervish and Donahue, 2019). Based on recent laboratory and

theoretical studies, which provide a nearly complete gas-phase oxidation mechanism for isoprene and

its major products (Wennberg et al., 2018, and references therein), recent SOA models have started to

employ explicit chemistry rather than relying on parameterization schemes, especially for isoprene

SOA. As a result, state-of-the-art chemical transport models have started to use a hybrid approach to

simulate SOA by using both parameterizations and explicit chemistry including heterogeneous reactions

(Pye et al., 2013; Marais et al., 2016; Jo et al., 2019; Shrivastava et al., 2019; Zare et al., 2019; Hodzic

et al., 2020).

When it comes to climate modeling studies, generally simplified SOA parameterizations have been

used (e.g. Sporre et al., 2019). In the fifth phase of the Coupled Model Intercomparison Project

(CMIP5), most models used a constant SOA yield without performing the oxidation of volatile organic

compounds (VOCs) and gas-to-particle partitioning calculations (Tsigaridis and Kanakidou, 2018).

Climate studies focusing on SOA employed more detailed schemes with initial oxidation of VOCs and

dynamic gas-to-particle partitioning such as the two-product scheme (Heald et al., 2008; Liao et al.,

2009; Wu et al., 2012; Cholakian et al., 2019a) and the VBS scheme (Megaritis et al., 2013; Pommier et

al., 2018; Zhang et al., 2018b). Recent studies used even more detailed processes such as SOA

formation from highly oxygenated molecules (Zhu et al., 2019) and heterogeneous reactions (Lin et al.,

2016).

A few previous studies examined the effects of different SOA schemes on future SOA prediction.

Cholakian et al. (2019b) showed that the complex VBS scheme shows a higher relative change of

biogenic SOA between future and historical simulations than the two-product scheme. On the other

hand, Day and Pandis (2015) reported that the inclusion of the volatility of primary particles and

chemical aging processes in the VBS scheme does not increase its sensitivity to climate change.

However, these studies only compared different volatility-based parameterizations, not including

explicit chemistry. As pointed out by Jo et al. (2019), both the SIMPLE and VBS parameterizations in

GEOS-Chem cannot capture the response to changes in $NO_x$ and $SO_2$ emissions predicted by the



explicit chemistry. This is critical because these emissions are expected to be substantially different in future climate/emissions scenarios (Feng et al., 2019a). Lin et al. (2016) used the hybrid approach to simulate global SOA using both parameterizations and explicit chemistry. In their sensitivity simulation, they investigated the effects of aerosol acidity on present-day and future SOA concentrations by comparing two cases using the constant and the pH-dependent reactive uptake coefficient of isoprene epoxydiols (IEPOX), and they concluded that the global average SOA production rate changes very little. This result is contrary to the results by Marais et al. (2017) who concluded that isoprene SOA mass yields per unit isoprene oxidized decreased from 13% in 1991 to 3.5% in 2013 over the Southeast US. This could be because Lin et al. (2016) investigated aerosol pH effects only, while keeping other conditions the same (e.g. $HO_2$/NO oxidation channel and aerosol surface area), whereas all the conditions were changed in Marais et al. (2017). Furthermore, Lin et al. (2016) calculated the effects of acidity on total SOA, but most of SOA components were not affected by the acidity change except for IEPOX-SOA. Stadtler et al. (2018) also investigated the IEPOX-SOA sensitivity to variable aerosol pH compared to fixed pH, and found 58% differences between the two cases.

In this study, we investigate the future SOA change as predicted by the Community Earth System Model version 2.1.0 (CESM2.1.0) using explicit IEPOX-SOA chemistry, and compare it with the SOA predictions by the VBS parameterization recently implemented in the model (Tilmes et al., 2019). We focus on IEPOX-SOA as we have a higher mechanistic understanding of the IEPOX-SOA formation compared to other SOA species, as well as IEPOX-SOA contributes a substantial mass fraction of SOA especially under low NO conditions (Hu et al., 2015). The CESM model includes several updates such as an adjustment of isoprene emissions, detailed isoprene chemistry, $NO_x$-dependent yields of the VBS, and the heterogeneous uptake of IEPOX. We evaluate the simulated IEPOX-SOA against aircraft campaign and surface network measurements. We then predict future IEPOX-SOA concentration changes for the mid and end of the 21st century under four shared socioeconomic pathways (SSPs) – SSP1-2.6, SSP2-4.5, SSP3-7.0, and SSP5-8.5 (O'Neill et al., 2016; Riahi et al., 2017). We also conduct sensitivity simulations to examine the effects of $CO_2$ inhibition on isoprene change and IEPOX-SOA



budget, and the effects of aerosol pH on IEPOX-SOA formation. Finally, we compare the explicit scheme and the VBS parameterization to investigate the differences between the two schemes in predicting future SOA concentrations.

## 2 Model description

### 2.1. General

We used CESM2.1.0 to simulate present and future climatic conditions (Danabasoglu et al., 2020). For the atmospheric configuration, the Community Atmosphere Model version 6 with comprehensive tropospheric and stratospheric chemistry representation (CAM6-chem) was used, with a horizontal resolution of 0.95° in latitude by 1.25° in longitude, and 32 vertical layers up to 1 hPa (40 km) (Emmons et al., 2020). Simulations were conducted for 10 years to construct multi-year averaged chemical fields under present (2005-2014) and future (2045-2054 and 2091-2100) atmospheric conditions, where the first two years were not included in the analysis (used for spin-up). Atmospheric (CAM6-chem) and land (Community Land Model version 5; CLM5) models were fully coupled online. We constrained sea surface temperatures and sea ice conditions from observations for present runs and from the Whole Atmosphere Community Climate Model version 6 (WACCM6) results for future runs. The WACCM6 results used in this study were those of the CMIP phase 6 (Gettelman et al., 2019).

For present-day conditions, anthropogenic emissions are from the Community Emissions Data System (CEDS) (Hoesly et al., 2018) and biomass burning emissions from the inventory developed for CMIP6 (van Marle et al., 2017). The SSPs were used for future climatic scenarios (Riahi et al., 2017). We selected all four Tier 1 scenarios – SSP5-8.5, SSP3-7.0, SSP2-4.5, and SSP1-2.6 (O'Neill et al., 2016; Gidden et al., 2019). Biogenic emissions were calculated online within CLM5 using the Model of Emissions of Gases and Aerosols from Nature (MEGAN) version 2.1 (Guenther et al., 2012). To address isoprene emission uncertainties related to $CO_2$ inhibition effects, we conducted two types of simulations with and without $CO_2$ inhibition effects (see Sect 2.2.3 for details). Aerosol simulations were based on the four-mode version of the Modal Aerosol Module (MAM4) (Liu et al., 2016), with



substantial changes to the treatments of inorganic and secondary organic aerosol species. Two very different treatments of secondary organic aerosol are used: an explicit treatment of IEPOX-SOA (Section 2.2.2), and a modified version of the Tilmes et al. (2019) 5-bin VBS SOA mechanism (Section 2.2.4). Dynamic partitioning of $H_2SO_4$, $HNO_3$, HCl, and $NH_3$ to each mode and the related particle-phase thermodynamics are calculated using the Model for Simulating Aerosol Interactions and Chemistry (MOSAIC) aerosol module (Zaveri et al., 2008; Zaveri et al., 2020; Lu et al., 2020).

For the model evaluation in Sect. 3, we used a specified dynamics option (FCSD compset) to reduce uncertainties related to dynamic simulations in the atmospheric model (CAM6). The meteorological fields (temperature, wind, and surface fluxes) were nudged towards the Modern-Era Retrospective analysis for Research and Applications version 2 (MERRA2) (Gelaro et al., 2017) with a relaxation time of 50 hours.

## 2.2. Model updates

### 2.2.1. Gas-phase chemistry of isoprene

Because the isoprene SOA yield strongly depends on the gas-phase chemistry, we used the MOZART-TS2 (Model of Ozone And Related chemical Tracers, Troposphere-Stratosphere V2) detailed isoprene chemical mechanism recently developed by Schwantes et al. (2020). The MOZART-TS2 chemical mechanism includes more comprehensive and updated both isoprene and terpene chemistry, but for computational efficiency, we only implemented the new isoprene chemistry. The updates for isoprene chemistry added 39 new compounds and 139 reactions on top of the MOZART-TS1 chemistry (Emmons et al., 2020), and includes updates to isoprene hydroxy hydroperoxide (ISOPOOH) and IEPOX chemistry relevant to this work. These updates were applied to all simulations in this work.

### 2.2.2. Heterogenous IEPOX reactive uptake

We implemented the heterogeneous uptake of IEPOX based on the work by Jo et al. (2019). We used the resistor model equation (Anttila et al., 2006; Gaston et al., 2014b) to calculate the reactive uptake coefficient of IEPOX ($\gamma_{IEPOX}$). The equation needs several parameters such as Henry's law constant and


diffusion coefficients of IEPOX in the aqueous core and in the organic layer, all input parameter values and equations are available in the supplement Sect. 1 of Jo et al. (2019). In addition to IEPOX-SOA, other SOAs were simulated using the VBS scheme as described in Sect. 2.2.4.

Aerosol pH was calculated online for each mode by the MOSAIC module (Zaveri et al., 2008), as implemented in CESM by Zaveri et al. (2020) and Lu et al. (2020). In addition, we modified MOSAIC to calculate submicron (Aitken and accumulation modes) aerosol pH without sea salt following Jo et al. (2019) based on previous studies showing that sea salt aerosols were dominantly externally mixed with submicron sulfate-nitrate-ammonium rather than internally mixed (Guo et al., 2016; Bondy et al., 2018;

Murphy et al., 2019). We note that sea salt was calculated in the model but excluded only for pH calculation. Removing sea salt resulted in substantially lower pH (more acidic aerosol) over the ocean, and a better agreement with the observationally-constrained pH values by Nault et al. (2020). The effects of sea salt on global aerosol pH and IEPOX-SOA formation are discussed in Sect. 4.2. It is worth noting that the submicron aerosol nitrate burden predicted by MAM4-MOSAIC is about 50% of

MAM7-MOSAIC (Lu et al., 2020; Zaveri et al., 2020), due to the different treatments of sub-micron sea salt and dust between MAM7 (Liu et al., 2012) and MAM4 (Liu et al., 2016), while the super-micron aerosol nitrate burdens are close. Nitrate aerosol affects pH and in turn, IEPOX-SOA formation, and pH also strongly affects nitrate aerosol formation. MAM4-MOSAIC has been found to predict similar nitrate levels compared to the observed nitrate aerosol concentrations over the US and East Asia (Lu et

al., 2020), but additional studies will be needed to evaluate the nitrate simulation in other regions.

    The effects of organic coatings have been often neglected in previous studies using the explicit chemistry for IEPOX-SOA (Lin et al., 2016; Marais et al., 2016; Budisulistiorini et al., 2017; Stadtler et al., 2018; Shrivastava et al., 2019), although it can reduce IEPOX-SOA formation rate by up to ~30% (Zhang et al., 2018a; Schmedding et al., 2019b). We considered the organic coating effect by assuming

inorganic-core and organic-shell mixture, as described in detail in Jo et al. (2019).

    The SOA yield of IEPOX reactive uptake was assumed to be 100% and IEPOX-SOA was treated as non-volatile in the model. This is consistent with other previous modeling studies (Marais et al., 2016;



Budisulistiorini et al., 2017; Stadtler et al., 2018; Schmedding et al., 2019b), based on field studies which showed that ambient IEPOX-SOA has very low volatility (Hu et al., 2016; Lopez-Hilfiker et al.,

2016; Riva et al., 2019). A recent chamber-based study by D'Ambro et al. (2019) confirmed the low-volatility of IEPOX-SOA, and suggested that the semivolatile products measured by some techniques in ambient IEPOX-SOA mostly resulted from thermal decomposition in those methods. On the other hand, they reported that the semi-volatile 2-methyltetrols that are also formed can evaporate after IEPOX reactive uptake, and can be lost to gas-phase reactions with OH and dry/wet deposition,

resulting in an IEPOX-SOA yield lower than unity. However, the evaporation is completed within one hour, and thus is not inconsistent with the very low volatility characteristics of ambient IEPOX-SOA. D'Ambro et al. (2019) also pointed out that the measured $\gamma_{IEPOX}$ is an order of magnitude or more higher than often used in models. We conducted sensitivity tests to investigate these uncertain parameters on model evaluation results in Sect. 3.

**2.2.3. Isoprene emission adjustment**

Isoprene emission changes are one of the main driving factors of IEPOX-SOA change in the future climate (Sect. 5). MEGAN v2.1 has been considered as the most reliable biogenic emission algorithm and used in many chemistry models (e.g. all models that participated in the intercomparison study by Hodzic et al. (2020) used MEGAN). However, there are still many uncertainties in calculating isoprene

emissions in models, especially for tropical forests which are less understood than their temperate and boreal counterparts due to great heterogeneity and many forest subtypes at scales of hundreds of meters (Batista et al., 2019). For example, ter Steege et al. (2013) reported that Amazonia harbors ~16,000 different tree species, and Silk et al. (2015) showed that there are 40,000-53,000 tree species in the tropics, in contrast to only 124 across temperate Europe. Furthermore, there have been fewer emission

studies over Tropics (< 20%) compared to their contribution to global biogenic VOC emissions (75%) (Guenther, 2013).

Satellite-based top-down studies have reported that chemistry models with the MEGAN v2.1 algorithm tend to overestimate isoprene emissions over the Tropics (Worden et al., 2019) including the





Amazon (Barkley et al., 2013), central Africa (Marais et al., 2012), and Borneo (Stavrakou et al., 2014). On the other hand, aircraft flux measurements suggested that MEGAN v2.0 and v2.1 underestimated isoprene emissions over the Amazon (v2.0 in Karl et al. (2007) and v2.1 in Gu et al. (2017)). Sarkar et al. (2020) showed that the same MEGAN algorithm with different inputs can both overestimate and underestimate the observed isoprene emission fluxes at a tower site over the Amazon, i.e. MEGAN v2.1 using emissions factors based on the plant functional type (PFT) overestimated the observed isoprene flux, whereas MEGAN v2.1 with the 1-km emission factor distribution map underestimated the flux.

In order to calculate the isoprene emissions as realistically as possible, especially for the Tropics, we used two constraints. First, we compared the simulated global isoprene emissions by CESM2.1.0/MEGANv2.1 (PFT-based emission factors, MODIS-based leaf area index (LAI)) with isoprene top-down estimates using HCHO from the Ozone Monitoring Instrument (OMI) (Bauwens et al., 2016). As discussed above, the model overestimated isoprene emissions especially for the Tropics (Sect. S1 and Figs. S1 and S2 in the Supplement). Second, we evaluated the modeled isoprene concentrations against aircraft measurements over the Amazon, which also suggested an overestimation (see Sect. 3.2 for details). To reduce discrepancies found in both of these comparisons, we reduced isoprene emission factors for tropical PFTs (broadleaf evergreen tropical tree and broadleaf deciduous tropical tree) by a factor of 2. The resulting isoprene emissions were comparable to OMI-based isoprene emission estimates in terms of the annual global total value, although there were still some regional differences (Figs. S1 and S2). For example, the base model overestimated OMI-based isoprene emissions over the Amazon by a factor of 3-4. Because satellite-based top-down emissions also have uncertainties in the inversion process (e.g., a priori source uncertainty), and the aircraft comparison of ambient isoprene over the Amazon (Sect. 3.2) only supported reducing isoprene emissions by a factor of 2, we considered that reducing isoprene emissions by a factor of 3-4 would have been excessive.

The $CO_2$ inhibition effect on isoprene emissions becomes important at high $CO_2$ concentrations under future climatic conditions, which can solely offset isoprene emission increase associated with the future temperature increase (Tai et al., 2013). Controlled chamber experiments revealed isoprene emissions are



lower at elevated $CO_2$ conditions (Wilkinson et al., 2009; Possell and Hewitt, 2011), and MEGANv2.1 includes the $CO_2$ inhibition effect in the isoprene emission calculation (Guenther et al., 2012). However, the quantitative effects of $CO_2$ inhibition on isoprene emission are still difficult to predict, likely due to the onset of complex feedback and feedforward biochemical mechanisms and interactive effects (Sharkey and Monson, 2017; Feng et al., 2019b). Bauwens et al. (2018) also reported the large

uncertainty associated with $CO_2$ inhibition calculations by using two distinct $CO_2$ inhibition parameterizations. Therefore, we conducted future simulations without $CO_2$ inhibition effects as a base case, and we additionally carried out simulations with $CO_2$ effects to estimate the robustness of our conclusions with regard to $CO_2$ effects. We note that the $CO_2$ inhibition effect is included in MEGANv2.1 in CESM as a default (Guenther et al., 2012).

## 2.2.4. $NO_x$-dependent yield of VBS scheme

The VBS scheme (Hodzic et al., 2016) as implemented in CESM2.1.0 considers only low $NO_x$ conditions (Tilmes et al., 2019). Because $NO_x$ levels in the future are predicted to change substantially (Feng et al., 2019a), it is important to consider $NO_x$-dependent yields to capture the response to $HO_2/NO$ ratio change in the future. We updated the VBS scheme to simulate high and low $NO_x$

pathways of SOA formation based on yields from Hodzic et al. (2016). For example, for isoprene, the default scheme includes a reaction with OH forming the SOA precursor gas phase species (SOAG) for the 5 VBS bins:

$$Isoprene + OH \rightarrow Isoprene + OH + \alpha_{low,0} * SOAG0 + \alpha_{low,1} * SOAG1 + \alpha_{low,2} * SOAG2$$

$$+ \alpha_{low,3} * SOAG3 + \alpha_{low,4} * SOAG4, \quad k = 2.54 \cdot 10^{-11} \times exp(\tfrac{410}{T}) \tag{1}$$

where SOAG0–4 represent gas-phase SOA species simulated by the VBS scheme, spanning saturation vapor concentrations from 0.01 $\mu$g m$^{-3}$ to 100 $\mu$g m$^{-3}$ at 300 K; $\alpha_{low,n}$ is the SOA molar yield at low $NO_x$ conditions for the volatility bin $n$, with values 0.0031, 0.0035, 0.0003, 0.0271, and 0.0474, respectively; $k$ is the reaction rate coefficient in cm$^3$ molecule$^{-1}$ s$^{-1}$, and T is the temperature in K.





To account for the high and low NOx conditions, we added two reactions to calculate the peroxy radical + $HO_2$ and peroxy radical + NO explicitly, as follows:

$$Isoprene + OH \rightarrow Isoprene + OH + \ ISOP\,O2V\,BS, \quad k = 2.7{\cdot}10^{-11} \times exp(\tfrac{390}{T}) \tag{2.1}$$

$$ISOP\,O2V\,BS + \ HO_2 \rightarrow HO_2 + \alpha_{low,0} * SOAG0 + \alpha_{low,1} * SOAG1 + \alpha_{low,2} * SOAG2$$
$$+ \alpha_{low,3} * SOAG3 + \alpha_{low,4} * SOAG4, \quad k = 2.12{\cdot}10^{-13} \times exp(\tfrac{1300}{T}) \tag{2.2}$$

$$ISOP\,O2V\,BS + NO \rightarrow NO + \alpha_{high,0} * SOAG0 + \alpha_{high,1} * SOAG1 + \alpha_{high,2} * SOAG2$$
$$+ \alpha_{high,3} * SOAG3 + \alpha_{high,4} * SOAG4, \quad k = 2.7{\cdot}10^{-12} \times exp(\tfrac{350}{T}) \tag{2.3}$$

Where $\alpha_{high,0\text{-}4}$ are 0.0003, 0.0003, 0.0073, 0.0057, and 0.0623, respectively. Isoprene, OH, $HO_2$, and NO are not consumed by the VBS reactions to avoid duplication with the detailed gas-phase isoprene chemistry in Sect. 2.2.1. In other words, the VBS doesn't affect gas-chemistry in the model, so all simulations in Table 1 have the same gas-chemistry. The isoprene peroxy radical (ISOPO2VBS) is a non-transported tracer to reduce computational cost. It is reasonable as a typical lifetime of isoprene peroxy radicals in the atmosphere is less than a few minutes (Wennberg et al., 2018; Jo et al., 2019). The reaction rate constant of isoprene + OH reaction matches the isoprene gas-phase reaction (Schwantes et al., 2020). Similar high NOx yields for SOA formed from terpenes, benzene, toluene and xylenes have been included in the mechanism.

## 3 Model evaluation

### 3.1. SEAC4RS

The Studies of Emissions, Atmospheric Composition, Clouds and Climate Coupling by Regional Surveys (SEAC[4]RS) campaign in August – September 2013 sampled the continental US troposphere with a heavy emphasis on the SE US and provided aircraft measurements of isoprene and its oxidation products (Marais et al., 2016; Toon et al., 2016). We evaluated the model against isoprene, isoprene hydroxy hydroperoxide (ISOPOOH), IEPOX, and IEPOX-SOA. Isoprene was observed by Proton Transfer Reaction Mass Spectrometry (PTR-MS) (de Gouw and Warneke, 2007; Müller et al., 2014).



Biomass burning impacted air masses ([CH3CN] > 200 ppt; Travis et al. (2016)) were excluded from the analysis in order to remove furan interference in detecting isoprene concentrations. The 2-methyl-3-buten-2-ol (MBO) interferences in the PTR-MS measurement of isoprene was assumed negligible over the Southeastern US. ISOPOOH and IEPOX were measured by chemical ionization mass spectrometer (CIMS) using the $CF_3O-$ ion, and IEPOX and ISOPOOH were individually quantified by tandem MS (Crounse et al., 2006; Paulot et al., 2009; St. Clair et al., 2010). IEPOX-SOA concentrations were quantified by applying positive matrix factorization (PMF) to aerosol mass spectrometer (AMS) OA measurements (Hu et al., 2015; Marais et al., 2016). We also excluded urban plumes as diagnosed by $[NO_2] > 4$ ppb and stratospheric air as diagnosed by $[O_3] / [CO] > 1.25$ mol mol$^{-1}$ based on Travis et al. (2016).

Figure 1 compares observed and modeled vertical profiles of isoprene, ISOPOOH, IEPOX, and IEPOX-SOA. The model generally captured vertical distributions but underestimated observed concentrations. Normalized mean biases (NMBs) were -64%, -73%, -52%, and -57% for isoprene, ISOPOOH, IEPOX, and IEPOX-SOA, respectively. The model substantially underestimated median values especially for isoprene and ISOPOOH but showed better performance for later generation products (IEPOX and IEPOX-SOA) with longer lifetimes. IEPOX-SOA concentrations over 6 km were overestimated by the model but underestimated below 6 km. This tendency is opposite to the vertical gradient of IEPOX-SOA over the Amazon in Sect. 3.2. In terms of frequency distributions (Figure S3), the model simulated more low concentration points and fewer high concentration points compared to measurements.

We conducted an additional sensitivity test as discussed in Sect. 2.2.2. We decreased the IEPOX-SOA yield from IEPOX to 0.2 (based on Fig. 8 of D'Ambro et al. (2019)) and increased the effective Henry's law constant (H*) by a factor of 5 ($8.5 \times 10^7$ M atm$^{-1}$ from $1.7 \times 10^7$ M atm$^{-1}$) to increase $\gamma_{IEPOX}$. The results are shown in Figs. S4 (a) and (d) for IEPOX and IEPOX-SOA, respectively. We found that IEPOX (NMB of -55%) stayed at similar levels while IEPOX-SOA (NMB of -87%) was much reduced in the model, which was attributed to the lower SOA yield of 20%, since the total production of IEPOX





stays the same. We further conducted sensitivity tests to investigate the IEPOX-SOA concentrations
using extremely high $H^*$ values (50 and 500 times higher, shown in Figs. S4 (b,c,e,f)). Even in these
extreme cases, IEPOX-SOA concentrations were substantially decreased (NMBs of -78% and -81%)
compared to the base case, due to low SOA yield. The mean modeled IEPOX and IEPOX-SOA ratios
between the extreme case ($H^*$ of $8.5 \times 10^9$ M atm$^{-1}$ and the yield of 0.2) and the base case ($H^*$ of $1.7 \times$
$10^7$ M atm$^{-1}$ and the yield of 1.0) were 0.68 and 0.51, which shows that the extreme case tends to
exacerbate the low bias in the model. Although the fraction of IEPOX taken up increases in this case, it
is not enough to compensate for the reduced yield compared to the base case.





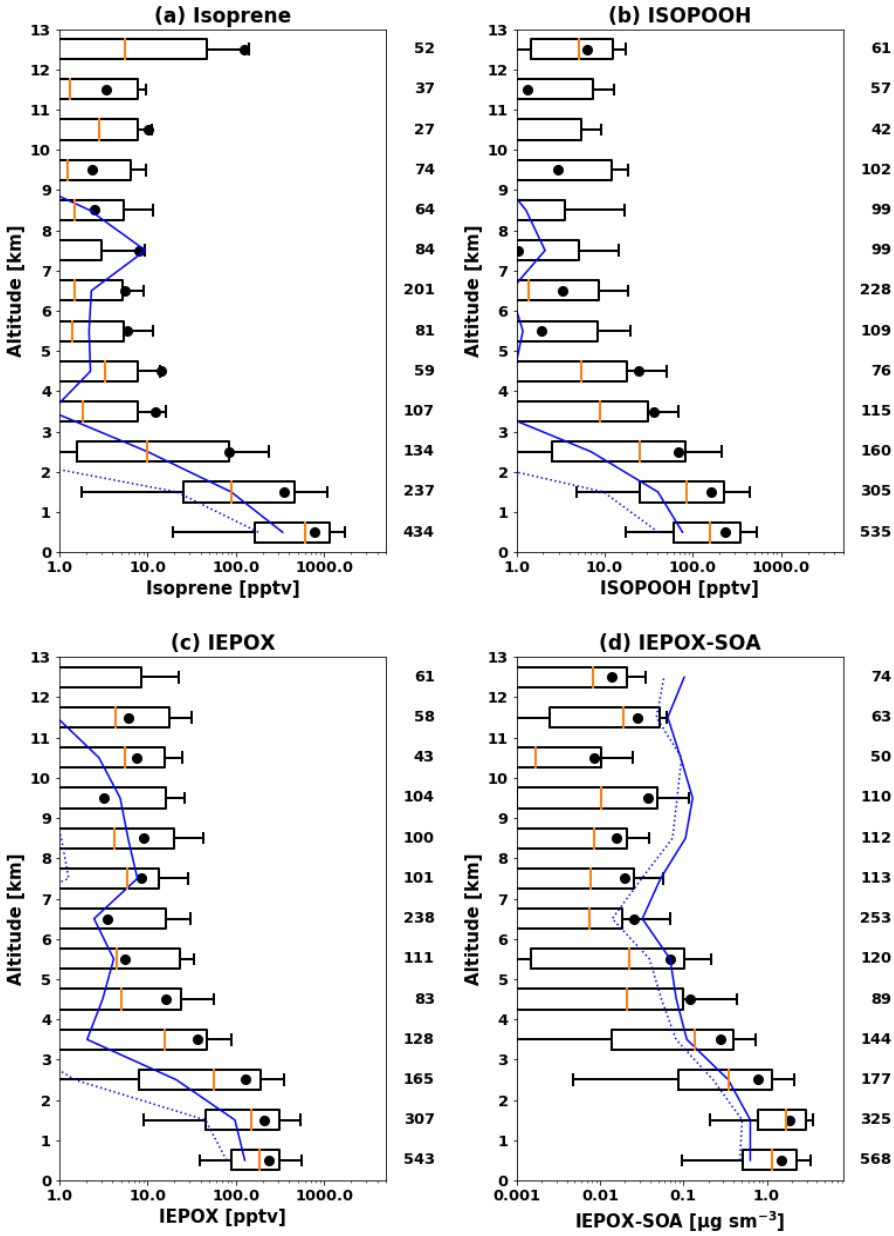

*Figure 1. Vertical profiles of (a) isoprene, (b) ISOPOOH, (c) IEPOX, and (d) IEPOX-SOA during the SEAC4RS campaign over the Southeast US. Profiles are binned to the 1 km vertical resolution grid. Boxes show the $25^{th} - 75^{th}$ percentile with the orange line being the median. Whiskers represent $10^{th}$ and $90^{th}$ percentiles and the black dot indicates the mean value. The model results are sampled along the flight tracks and shown in blue, with the solid line for mean and the dashed line for median values. The number of 60 s merged data points in each km interval is shown at the right of each panel.*



## 3.2. GoAmazon and ACRIDICON-CHUVA

The Green Ocean Amazon (GoAmazon) experiment provided a variety of chemical measurements over the Amazon with aircraft and surface measurements in wet and dry seasons of 2014. One of the main goals of the GoAmazon campaign was to investigate the interaction between pollution plumes from Manaus city and background conditions of the central Amazon basin. There were two Intensive Operating Periods (IOP1 and IOP2). IOP1 was carried out from 1 February to 31 March 2014 in the wet

season, and IOP2 was conducted from 15 August to 15 October 2014 in the dry season (Martin et al., 2016). A low-flying G-159 Gulfstream I (G-1) aircraft collected data including isoprene mostly in the atmospheric boundary layer during IOP-1 and -2. The German High Altitude and Long Range Research Aircraft (HALO) was deployed during the ACRIDICON-CHUVA (Aerosol, Cloud, Precipitation, and Radiation Interactions and Dynamics of Convective Cloud Systems) field campaign, which was

performed in cooperation with IOP2 of the GoAmazon campaign (Wendisch et al., 2016). Unlike the G-1 aircraft, HALO flew not only in the boundary layer but also in the mid- and upper troposphere. Isoprene was measured by PTR-MS (Shilling et al., 2018) in the G-1 and IEPOX-SOA was calculated from AMS data using PMF analysis on the G-1 (Shilling et al., 2018) and on HALO (Schulz et al., 2018) using the *m/z* 82 tracer method by Hu et al. (2015).

This dataset was used to evaluate the model results in typical tropical forest conditions characterized by elevated isoprene emissions. Given the coarse model resolution, we have filtered out urban and biomass burning plumes from low altitude G-1 flights, and focused on evaluating the model's ability to simulate SOA in background conditions. Based on cluster analyses at the surface (T3) site by de Sá et al. (2018) for IOP1 and de Sá et al. (2019) for IOP2, we used the following criteria for removing

polluted air masses from the data for evaluation: $NO_y$ > 1ppb, ozone > 20 ppb, and particle number > 1200 $cm^{-3}$ for IOP1 and $NO_y$ > 1.3 ppb, ozone > 36 ppb, and particle number > 2240 $cm^{-3}$ for IOP2. We applied these criteria to observations from the G-1 aircraft, by assuming that both the G-1 aircraft and the T3 surface site were affected by similar air masses. The G-1 aircraft mostly sampled the boundary layer (~75% data points below 1 km, see Fig. 2), and its flight tracks were close to the T3 site (Martin et




al., 2016). To estimate the impact of the uncertainty in modeled isoprene emissions, we conducted three simulations with different isoprene emissions by scaling the emission factors of Tropical trees by 100% (base), 50% (half), and 25% (quarter).

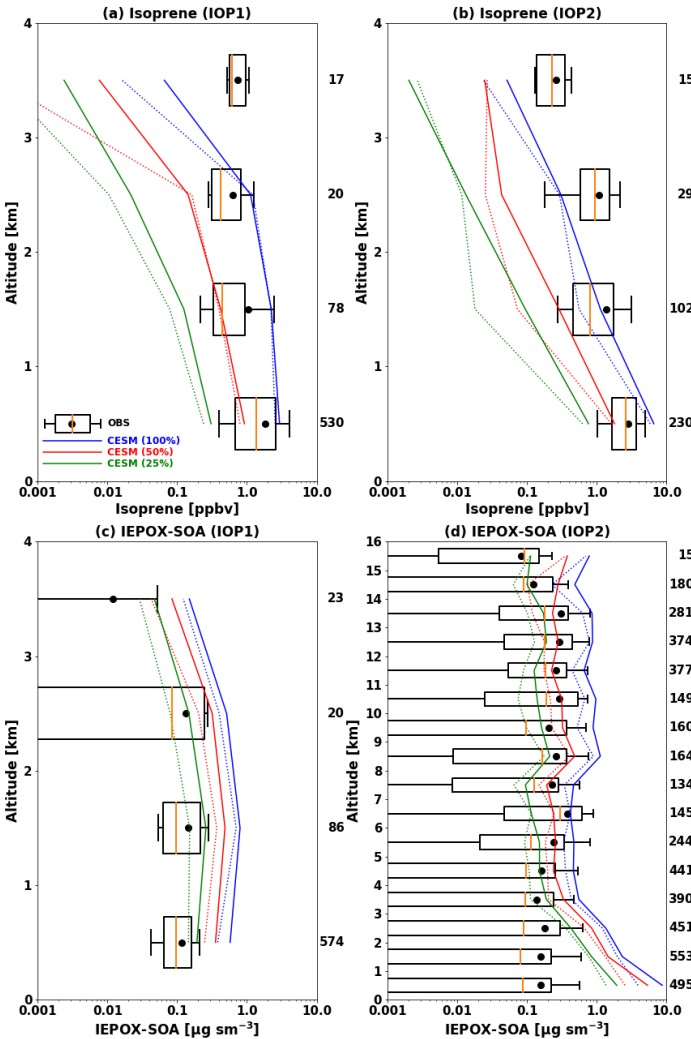

*Figure 2. Vertical profiles of isoprene (a,b) and IEPOX-SOA (c,d) during the GoAmazon (a,b,c) and*
*ACRIDICON-CHUVA (d) campaigns. Profiles are binned to the 1 km vertical resolution. Boxes show the $25^{th} - 75^{th}$ percentile with the orange line being the median. Whiskers represent the 10th and $90^{th}$ percentiles and the black dot indicates the mean value. The model results are shown in blue, red, and green for the base, half, and quarter isoprene emissions from Tropical tree PFTs, respectively. The solid line represents mean and the dashed line indicates median values. The number of data points in each km*
*interval is shown at the right of each panel.*





Figure 2 shows vertical profiles of isoprene and IEPOX-SOA from the G-1 (a,b,c) and HALO (d), respectively. Isoprene and IEPOX-SOA concentrations simulated by CESM2.1.0 with MEGANv2.1 substantially overestimate observed concentrations. The data in Fig. 2 shows NMBs of 65% and 99% for isoprene during IOP1 (a) and IOP2 (b), respectively. The overestimations were higher for IEPOX-SOA, which were 430% during IOP1 (c) and 769% during IOP2(c). We attribute these overestimations to too high isoprene emissions as discussed in Sect. 2.2.3. A sensitivity simulation with the isoprene emissions from tropical trees decreased by 50% reduced model biases to -51% and -46% for isoprene during IOP1 and IOP2, respectively, and to 225% and 394% for IEPOX-SOA. A further reduction of isoprene emissions to 25% led to substantial underprediction of isoprene concentrations with biases of -84% during IOP1 and -77% during IOP2. Mean IEPOX-SOA concentrations were still overestimated (75% and 121%), with strong variability in biases according to altitude (Fig. 2d). Even the model with the lowest isoprene emissions overestimated IEPOX-SOA below 5 km, but it substantially underestimated IEPOX-SOA above 5 km (green line in Fig. 2d). One possible reason are inaccuracies in the convection scheme in the model, because the ACRIDICON-CHUVA period was strongly influenced by tropical deep convection (Schulz et al., 2018). It is worth noting that the convection scheme in CAM6 (CLUBB; Cloud Layers Unified By Binormals) has shown improved performance for tropical deep convection over land compared to the previous convection schemes in CESM (Danabasoglu et al., 2020), but it may still contribute to the observed differences. In terms of IEPOX-SOA concentrations above 5 km, NMBs were 181%, 6%, and -42% for the base, half, and quarter isoprene emission cases, respectively.

The higher model IEPOX-SOA bias against the ACRIDICON-CHUVA (121–769%) compared to the GoAmazon (75–430%) measurements can be explained by particle losses in the constant pressure inlet used by the HALO AMS. Mei et al. (2020) reported that there were particle losses in the constant pressure inlet used on the HALO AMS, by comparing G1 AMS and HALO AMS during coordinated flights. The differences between G1 and HALO AMS were up to a factor of two below 2 km (Fig. 11 in Mei et al. (2020)).



We further investigated the effects of IEPOX-SOA evaporation (Sect. 2.2.2) on the model performance, by using H* of $8.5 \times 10^7$ M atm$^{-1}$ and the yield of 0.2 (IEPOX-SOA yield from IEPOX reactive uptake), especially for the model with the half isoprene emissions (Fig. S5). NMBs were 11% and 28% over all altitudes for IOP1 and IOP2, respectively. For the ACRIDICON-CHUVA campaign (Fig. S5d), this improvement results from a combination of an overestimation of IEPOX-SOA below 3 km and an underestimation above 3 km. The model showed a strong vertical gradient which was different from the observed vertical profiles. NMB was -70% above 5 km. However in general, contrary to the evaluation against SEAC4RS measurements (Sect. 3.1), the inclusion of the recent findings from the chamber study (D'Ambro et al., 2019) improved the model simulation. One possible explanation is different IEPOX-SOA molecular components between the Southeast US and the Amazon, however the reported composition is similar for both locations (Yee et al., 2020). This new approach also changed IEPOX as shown in Sect 3.1. Combined field measurements of IEPOX and IEPOX-SOA, and a more detailed molecular-level scheme for the IEPOX-SOA model will be needed to clarify the behaviors of IEPOX-SOA in different regions.

Considering the isoprene and IEPOX-SOA evaluations together, the model with the half isoprene emissions showed relatively better performance, although it overestimated IEPOX-SOA concentrations below 5 km. The other model results showed too high isoprene and IEPOX-SOA (base emission case) or too low isoprene mixing ratio (quarter emission case). The model with the half isoprene emission also showed better results in terms of the frequency distribution (Fig. S6). Hereafter, the model simulations are based on the reduced isoprene emission factors by half for Tropical tree PFTs. This treatment mainly affected isoprene emissions over the Tropics (25°S – 25°N, see Fig. S1e).

### 3.3. Global surface AMS dataset

We evaluated the model against global IEPOX-SOA concentrations reported by Hu et al. (2015), who quantified IEPOX-SOA by applying PMF to AMS measurements. They reported surface IEPOX-SOA concentrations averaged for each campaign period (from a month to a few months) as shown in Table S1. Figure 3 shows the comparison of modeled versus observed IEPOX-SOA concentrations at the





surface spanning years between 2008 and 2014. In this section, the modeled isoprene emissions of tropical trees were reduced by half according to the evaluation in Sect 3.2. Analogous to the evaluations

against aircraft measurements above, the model generally underestimated IEPOX-SOA over the southeast US but overestimates IEPOX-SOA over Amazonia. However, the model captured reasonably well the spatial variability across several chemical regimes as well as the interannual variability of IEPOX-SOA ($R^2 = 0.65$).

These evaluations show the challenges in accurately predicting isoprene and IEPOX-SOA. The

model underestimated IEPOX-SOA during SEAC$^4$RS (Fig. 1) but overestimated IEPOX-SOA in the Amazon (Fig. 2). Furthermore, the model biases as a function of altitude over the Amazon were opposite to those over the southeast US during SEAC$^4$RS. Even within the southeast US region, the model showed different performance for different years (Centreville, Look Rock, Atlanta in Fig. 3). The apparent differences in model performance across regions, altitudes, and time periods, indicate that the

significant complexities still exist in the ambient atmosphere for IEPOX-SOA formation, despite significant advances in the laboratory studies. For instance, results cannot be consistently improved by adjusting single parameters like Henry's law constant, and other model processes such as vertical mixing and wet deposition may be imperfect as well. Although we implemented a new comprehensive gas-phase and heterogeneous chemistry, more studies are needed to reduce the gap between

observations and models, such as the IEPOX-SOA evaporation and molecular characteristics (D'Ambro et al., 2019), aerosol pH biases in chemical transport models (Nault et al., 2020), and large uncertainties in organic coating effects and phase separation calculations (Schmedding et al., 2019a).

We have selected the model version evaluated in this section to assess IEPOX-SOA concentrations under future climatic scenarios. This version has the most comprehensive representation of gas-phase

chemistry of isoprene and IEPOX heterogeneous uptake that is available up to date, and it captures reasonably well the measured IEPOX-SOA concentrations across different chemical regimes. The mechanistic approach included with the explicit chemistry in this model can assess the sensitivity of




IEPOX-SOA to multiple factors, which cannot be evaluated from widely used parameterizations such as
the two product scheme or VBS approaches, as shown below.


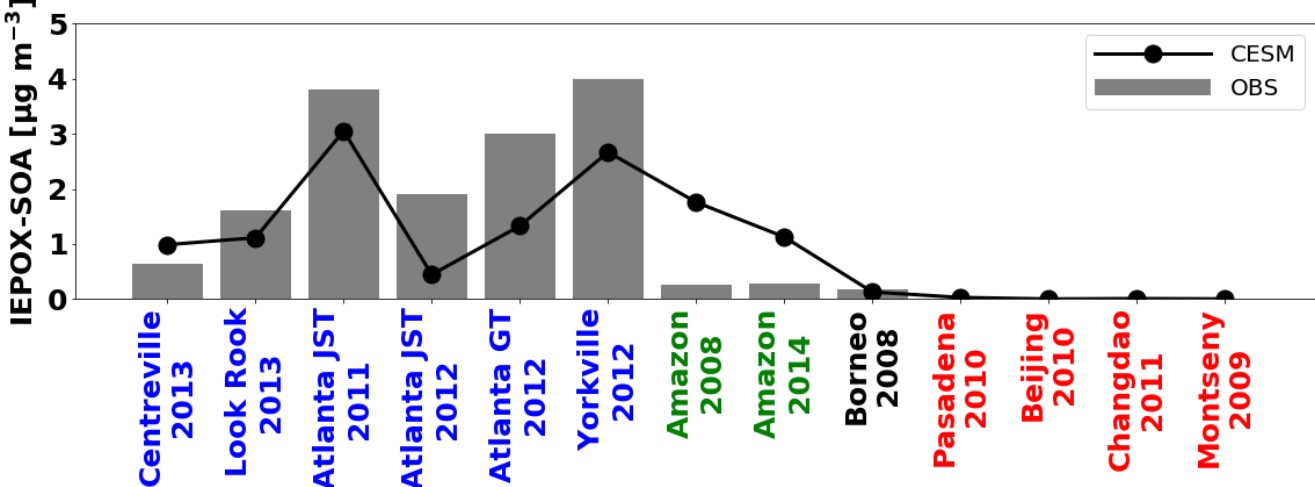

*Figure 3. Global surface IEPOX-SOA concentrations calculated by Hu et al. (2015) (gray bar) versus
the model results (solid line). Detailed information about site locations, time period, and IEPOX-SOA
concentrations are available in Table S1. The southeast US, the Amazon, and urban regions are shown
in blue, green, and red, respectively.*





Table 1. Description of IEPOX-SOA simulations under present and future conditions.

| Simulation name | SOA scheme | Years[1] | Sea salt in aerosol pH calculation | $CO_2$ inhibition effect in isoprene emissions |
|---|---|---|---|---|
| EXP_2010 | Explicit | 2005-2014 | Excluded | NO |
| EXP_CO2_2010 | Explicit | 2005-2014 | Excluded | YES |
| EXP_SS_2010 | Explicit | 2005-2014 | Included | NO |
| EXP_SS_CO2_2010 | Explicit | 2005-2014 | Included | YES |
| EXP_2050_SSPx[3] | Explicit | 2045-2054 | Excluded | NO |
| EXP_CO2_2050_SSPx[3] | Explicit | 2045-2054 | Excluded | YES |
| EXP_2090_SSPx[3] | Explicit | 2091-2100 | Excluded | NO |
| EXP_CO2_2090_SSPx[3] | Explicit | 2091-2100 | Excluded | YES |
| EXP_SS_2090_SSPx[3] | Explicit | 2091-2100 | Included | NO |
| EXP_SS_CO2_2090_SSPx[3] | Explicit | 2091-2100 | Included | YES |
| VBS_2010 | VBS | 2005-2014 | N/A[2] | NO |
| VBS_CO2_2010 | VBS | 2005-2014 | N/A[2] | YES |
| VBS_2050_SSPx[3] | VBS | 2045-2054 | N/A[2] | NO |
| VBS_CO2_2050_SSPx[3] | VBS | 2045-2054 | N/A[2] | YES |
| VBS_2090_SSPx[3] | VBS | 2091-2100 | N/A[2] | NO |
| VBS_CO2_2090_SSPx[3] | VBS | 2091-2100 | N/A[2] | YES |

[1] First two years were not used for analysis.

[2] SOA simulated by the VBS does not have a dependency on aerosol pH.

[3] Here x can be 5 (SSP5-8.5), 3 (SSP3-7.0), 2 (SSP2-4.5), and 1 (SSP1-2.6).



## 4 IEPOX-SOA changes in future climatic conditions

In this section, we present future IEPOX-SOA concentrations predicted by the explicit chemistry using four SSP scenarios (SSP5-8.5, SSP3-7.0, SSP2-4.5, and SSP1-2.6), along with IEPOX-SOA under present conditions. The characteristics of the simulations are listed in Table 1. One present (2010s) and two future (2050s and 2090s) simulations were conducted. In each set of simulations shown in Table 1, we simulated 10 years to account for interannual variability. The first 2 years were discarded as a spin-up period, and the remaining 8 years were used for the analysis. In addition to the future IEPOX-SOA changes, we investigated the effects of aerosol pH (Sect. 4.2) and $CO_2$ inhibition of isoprene emissions (Sect. 4.3) on IEPOX-SOA budgets.

### 4.1. IEPOX-SOA changes

Figure 4 shows global mean surface concentrations of IEPOX-SOA under present and future conditions for the explicit (EXP) simulations, while Figure 5 shows IEPOX-SOA changes under future climates with respect to present conditions. Generally, IEPOX-SOA concentrations are predicted to increase globally, especially under high climate forcing scenarios (SSP3-7.0 and SSP5-8.5). The driving factor for increased IEPOX-SOA concentrations is the increase in isoprene emissions, as shown in Fig. 6. Higher surface temperatures mainly increased isoprene future emissions under the base case simulations without $CO_2$ inhibition (see Sect. 4.3 for further discussion of multiple factors changing isoprene emissions under future climatic scenarios). Global mean surface temperatures were 278.8, 280.7 and 285.3 K for EXP_2010, EXP_2090_SSP1, and EXP_2090_SSP5 simulations, respectively.

The westward continental outflow from the Amazon was enhanced under all SSP scenarios as shown in Figs. 4 and 5. This was due to longer lifetimes of isoprene and its products (Fig. 6), which resulted from lower OH concentrations in future conditions (Fig. S7). Increased OH consumption by higher isoprene and decreased OH production by less $NO_x$ (Fig. S8) led to lower OH concentrations.



*Figure 4. Global mean IEPOX-SOA concentrations at the surface simulated in (a) EXP_2010, (b-e) EXP_2050_SSPx, and (f-i) EXP_2090_SSPx.*



Figure 5. Projected changes in global mean surface IEPOX-SOA concentrations from present (EXP_2010) conditions to (a-d) 2050s (EXP_2050_SSPx) and (e-h) 2090s (EXP_2090_SSPx) conditions.





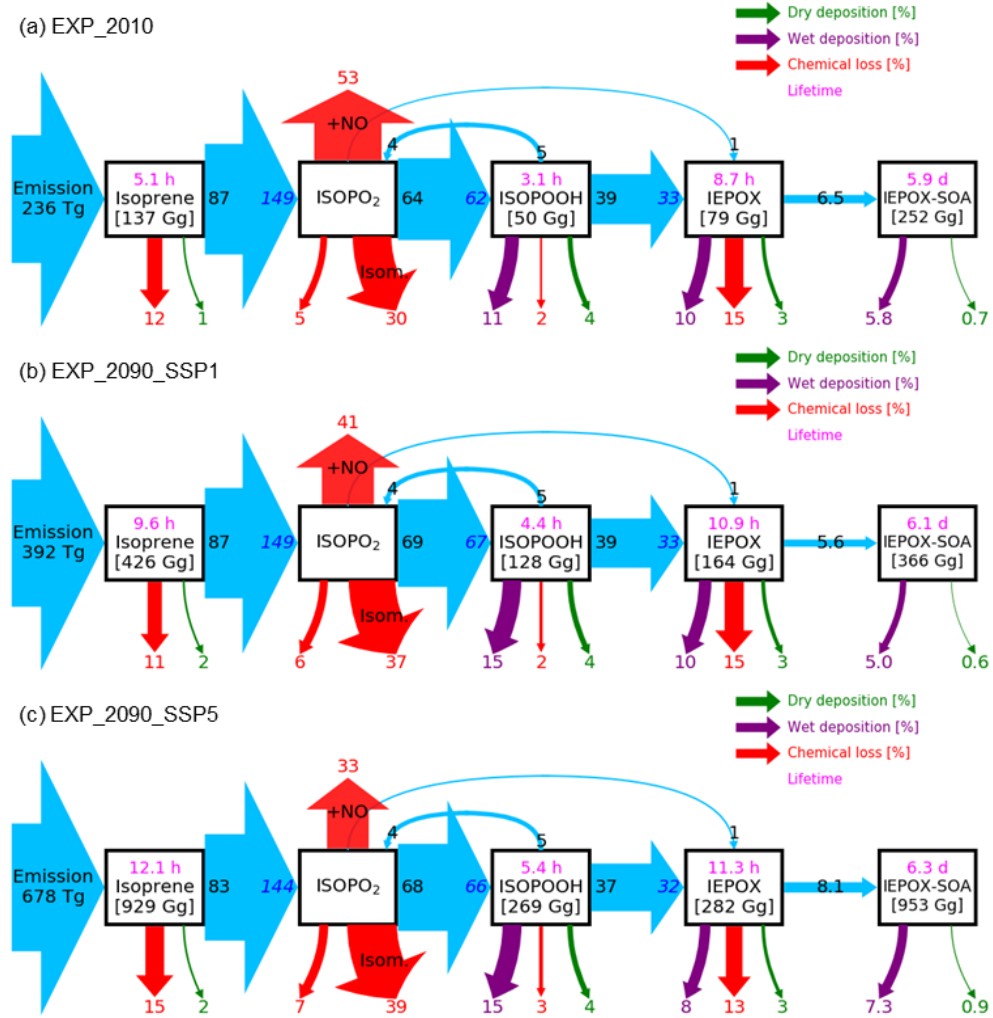

*Figure 6. Relative mass flux diagrams for global IEPOX-SOA budget analysis in CESM2.1.0. Blue arrows represent the IEPOX-SOA formation pathway. Mass fluxes (%) are calculated relative to isoprene emission for each scenario (i.e. normalized to the isoprene emission) and shown outside the boxes. Absolute values of mass fluxes in Tg yr$^{-1}$ are presented in Fig. S9. Burdens (in Gg) are shown inside. Two flux numbers are shown if the loss amount of reactant (black number) differs from the production amount of product (blue number), which is caused by the different molecular weights and product yields. The IEPOX formation initiated from isoprene + NO$_3$ reaction is not explicitly shown, and its amount is added to the IEPOX formation from ISOPO$_2$ + NO pathway. Red arrows show the chemical pathways not leading to IEPOX-SOA formation. For isoprene peroxy radical (ISOPO$_2$), the chemical loss is separated into three categories: the reaction with NO (+NO), isomerization (Isom.), and others. Purple and green arrows indicate wet and dry depositions, respectively. Global tropospheric burdens are shown in brackets for transported species in the model. Lifetime is shown in pink.*





The fate of the isoprene peroxy radical also changed, as the relative contribution of the reaction with NO and isomerization was quite different between the present and future conditions (Fig. 6). In present conditions, the reaction with NO (53% of the isoprene emission amount) was more dominant than the isomerization pathway (30%). However, it was reversed under the SSP5-8.5 scenario, with the flux of the isomerization pathway (39%) being higher than that of the reaction with NO (33%). There were two main causes: (1) isomerization rates increase with temperature (Bianchi et al., 2019), and (2) the global mean $NO_x$ emissions decrease under all future SSP scenarios (Turnock et al., 2020).

Interestingly, the fraction of isoprene emissions leading to ISOPOOH formation did not change much (62–67%), although the absolute amount was increased due to the isoprene emissions increase (Fig. S9). The IEPOX formation rates from isoprene emissions were nearly constant (32–33%) for present and all SSP scenarios (SSP2-4.5 and SSP3-7.0 are not shown), due to compensating effects between factors discussed above.

The IEPOX-SOA yield from isoprene emissions varied from 5.6% (SSP1-2.6) to 8.6% (SSP3-7.0), although IEPOX/isoprene flux ratios were almost the same. Much of this difference is due to predicted changes in sulfate aerosol concentrations over the Amazon (Fig. S10). Higher sulfate concentrations over the Amazon not only increased the available surface area where IEPOX reactive uptake takes place but also enhanced aerosol acidity (reduced aerosol pH, Fig. S11), which led to a faster formation of IEPOX-SOA.

These findings show that IEPOX-SOA formation is nonlinear. Thus, the formation cannot be easily simplified as various processes are involved, such as isoprene emissions affecting OH consumption and thus the lifetime of other chemical species, NO and temperature affecting the fate of peroxy radicals, and aerosol surface area and pH affecting IEPOX reactive uptake. In the following section (Sect. 5), we present a comparison between the explicit chemistry and the VBS approach to investigate the ability of an empirical parameterization that is typical of those currently used in most models to simulate SOA under future climates.



## 4.2. Effects of aerosol pH on IEPOX-SOA formation

Here we discuss the effects of aerosol pH on IEPOX-SOA formation, by comparing EXP and EXP_SS simulations. Figure 7 shows IEPOX-SOA concentrations and ratios between future and present conditions simulated by EXP_SS which includes sea salt aerosols in the aerosol pH calculations. The
545 predicted IEPOX-SOA levels were lower in the EXP_SS than in the EXP case, due to higher aerosol pH which hindered acid-catalyzed reactions of IEPOX (Gaston et al., 2014a). Figure 8 illustrates that IEPOX-SOA formations were decreased by 25–37% in the EXP_SS simulation due to less acidic aerosols, even though emissions, chemistry, and deposition fluxes leading to IEPOX were almost the same compared to EXP. This result indicates that the correct treatment of inorganic aerosols and related
thermodynamic calculations are critically needed for the accurate simulation of organic aerosols. The different pH treatments could also impact the nitrate formation, the accumulation mode nitrate burden was reduced by 19% in EXP_SS (14.8 GgN) compared to EXP (18.2 GgN).

## 4.3. Isoprene emission and IEPOX-SOA changes with $CO_2$ inhibition effects

As discussed in Sect. 2.2.3, the $CO_2$ inhibition effect on isoprene emissions is especially important for
predicting future climate, but large uncertainties exist. We examined the predicted IEPOX-SOA concentrations with $CO_2$ inhibition (EXP_$CO_2$) and compared to the cases without $CO_2$ inhibition (EXP). Predicted global mean surface IEPOX-SOA concentrations in present and future (2090s) conditions are presented in Fig. 9 (a-e) along with relative changes in the future compared to the present conditions (f-i). While aerosol pH affected IEPOX-SOA in all the simulations including present
conditions (Sect. 4.2), $CO_2$ inhibition played a critical role only in future simulations with higher $CO_2$ concentrations. As shown in Fig. 10, global isoprene emissions in EXP_CO2_2090 simulations were substantially lower than in EXP_2090 simulations (Fig. 6), especially for the SSP5-8.5 scenario with high $CO_2$ (53% decrease). Consequently, the annual IEPOX-SOA formation fluxes were also reduced, 24.7 Tg yr$^{-1}$ (EXP_2090_CO2) from 55.0 Tg yr$^{-1}$ (EXP_2090) for the SSP5-8.5 scenario. Global surface





IEPOX-SOA concentrations were also decreased due to $CO_2$ inhibition across all SSP scenarios with

higher relative changes for SSP3-7.0 and SSP5-8.5 (Fig. S12).

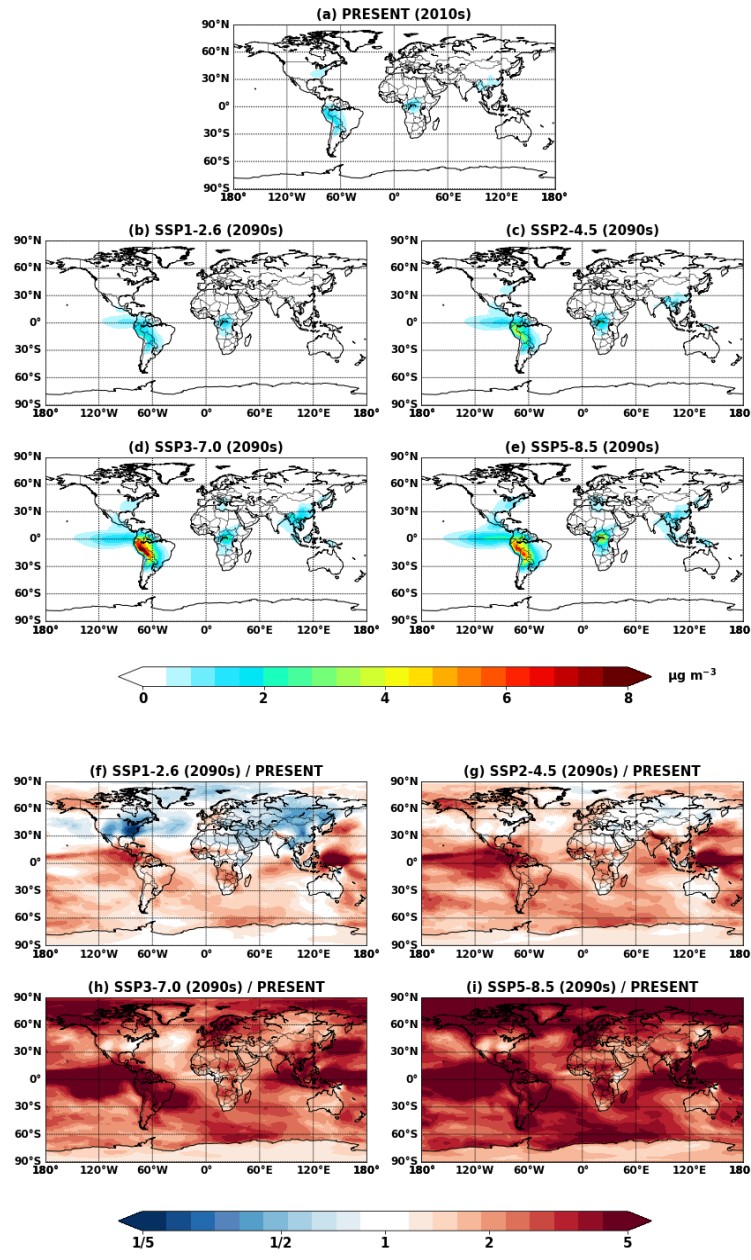

*Figure 7. Global mean IEPOX-SOA concentrations at the surface simulated in (a) Explicit_SS_2010 and (b-e) Explicit_SS_2090_SSPx. (f-i) Ratios of IEPOX-SOA between future (b-e) and present (a)*
570 *conditions.*





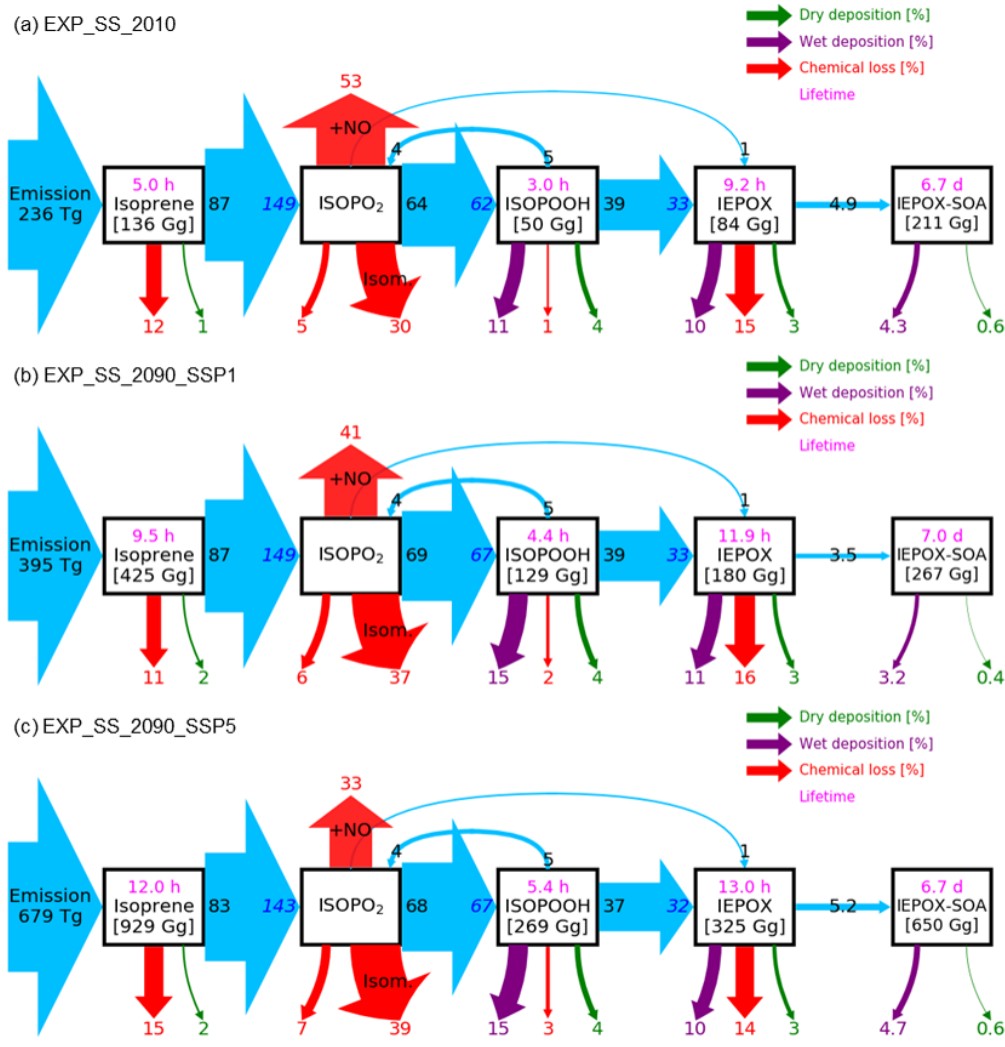

*Figure 8. Same as Fig. 6 but for EXP_SS simulations (including sea salt in aerosol pH calculation).*





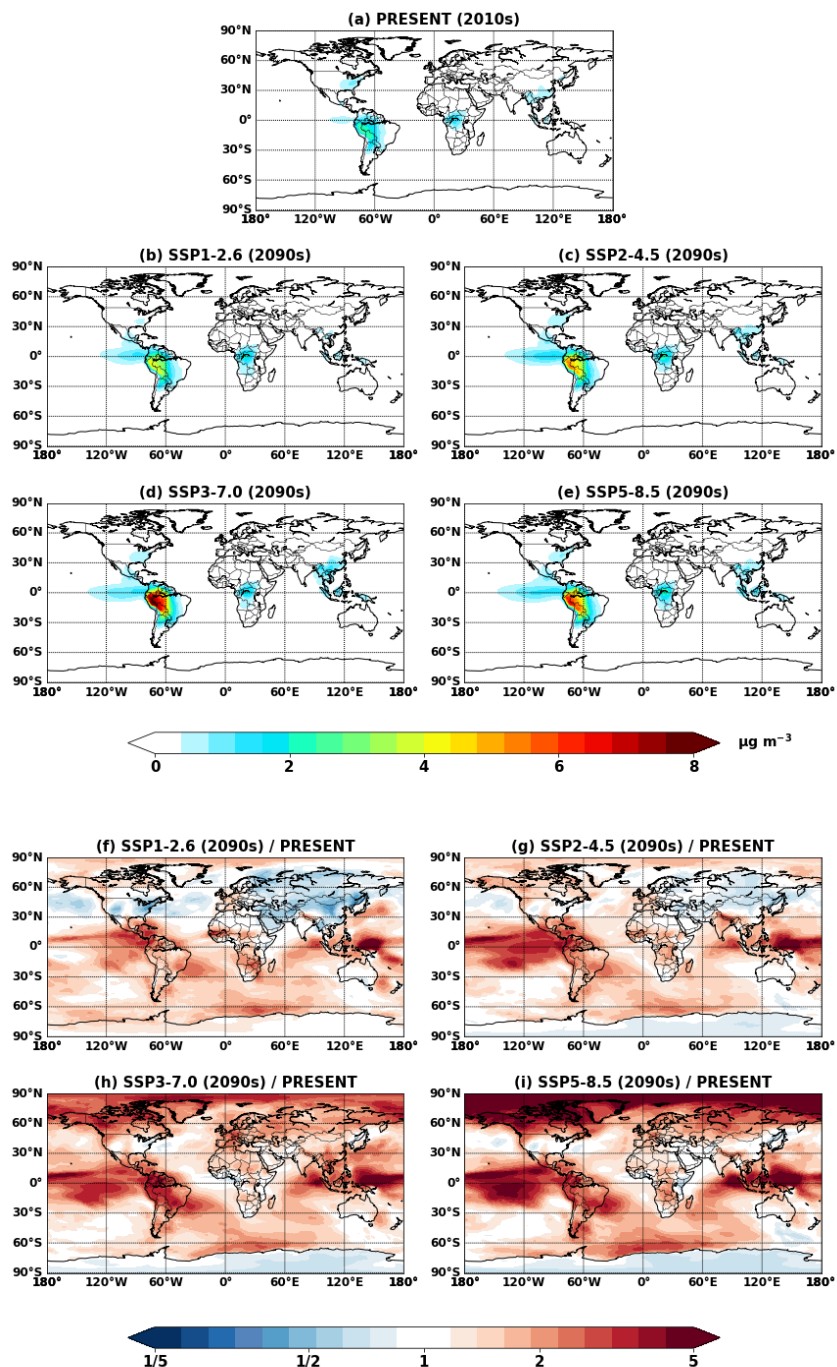

*Figure 9. Global mean IEPOX-SOA concentrations at the surface simulated in (a) Explicit_CO2_2010 and (b-e) Explicit_CO2_2090_SSPx. (f-i) Ratios of IEPOX-SOA between future (b-e) and present (a) conditions.*



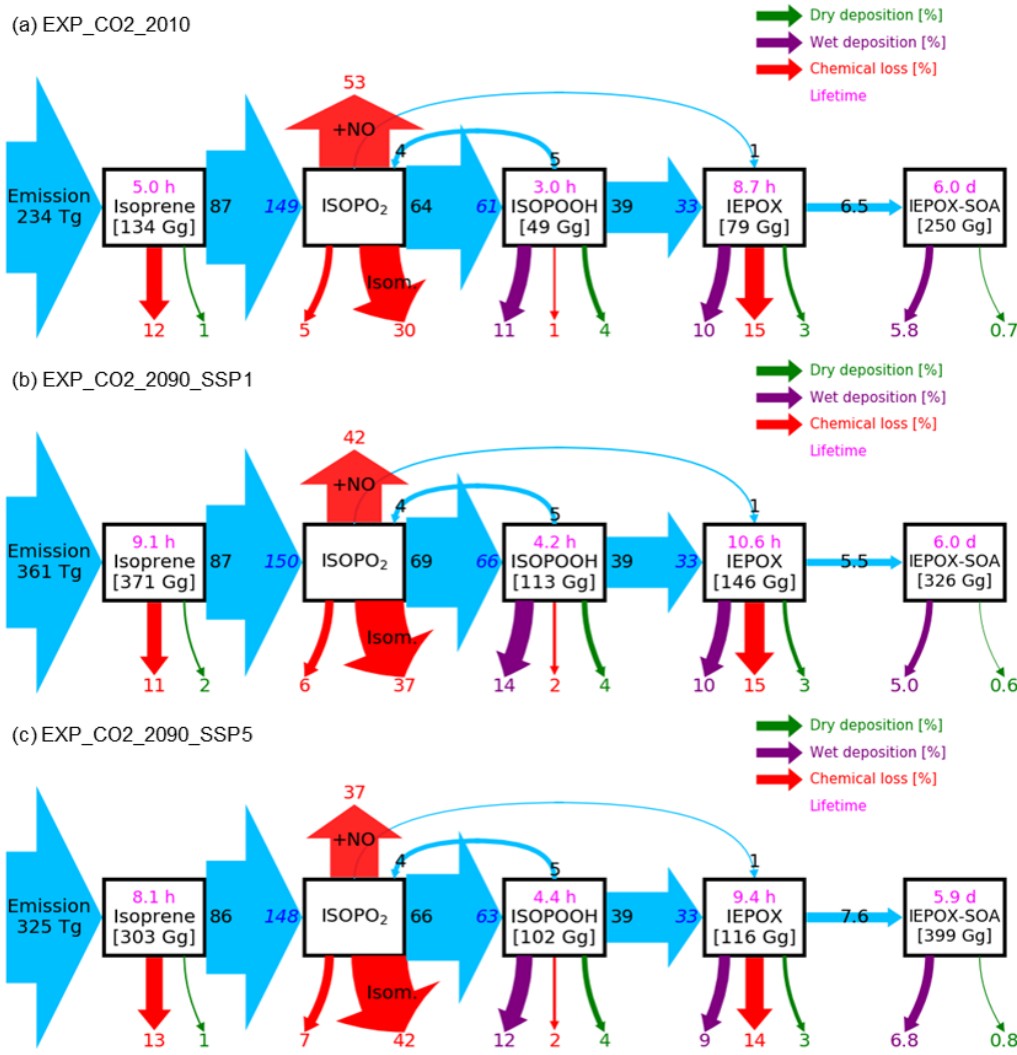

*Figure 10. Same as Fig. 6 but for EXP_CO2 simulations (CO$_2$ inhibition effect is taken into account).*





Isoprene emission calculations by MEGANv2.1 are based on a simple mechanistic model that considers the major processes driving variations by using activity factors (Guenther et al., 2012). We broke down activity factor changes under SSP1-2.6 (Fig. S13) and SSP5-8.5 (Fig. S14) with respect to present conditions. The main driving factor in the isoprene emission increase was the response to temperature, although both LAI and light factors also contributed to the emission increase. On the other hand, $CO_2$ inhibition counteracted the temperature effect, surpassing it especially for the SSP5-8.5 scenario.

Isoprene emission changes due to $CO_2$ inhibition also affected chemistry. Lifetimes of IEPOX-SOA precursors were decreased drastically under the SSP5-8.5 scenario when $CO_2$ inhibition was included in the isoprene emission calculation (EXP_CO2, Fig. 10). This was due to higher OH concentrations in EXP_CO2, caused by less OH consumption by isoprene and its products (Fig. S15). The change increased as the forcing ($CO_2$) increased. It also affected the relative changes in IEPOX-SOA concentrations between future and present conditions, which has important implications for high forcing scenarios (SSP3-7.0 and SSP5-8.5).



## 5 Comparison with the VBS approach

In this section, we examined isoprene SOA and its future changes by the VBS approach (VBS case)
and compared it against IEPOX-SOA by the explicit chemistry (EXP case). The VBS approach only
simulates total isoprene SOA and does not separate IEPOX-SOA (Tilmes et al., 2019). Because
IEPOX-SOA is mostly formed by the isoprene + OH reaction pathway, here we only considered
isoprene SOA formed  via the isoprene + OH VBS reaction pathway. The VBS SOA from isoprene + $O_3$
and isoprene + $NO_3$ pathways were excluded in the analysis, although they were included in the model
simulation. Hereafter, we refer to isoprene SOA as the VBS SOA from isoprene + OH pathway
simulated by the VBS, in order to distinguish it from IEPOX-SOA simulated by the explicit chemistry.

Figure 11 shows the VBS SOA and its change in the future. The VBS scheme captured global spatial
patterns of isoprene SOA although absolute magnitudes were lower than IEPOX-SOA. VBS SOA was
generally predicted to be increased under the future climate, which was also analogous to the
IEPOX-SOA change (Fig. 5). However, the concentration changes in the future were regionally
different. The model with the explicit chemistry predicted the IEPOX-SOA concentration decrease in
the 2090s under the SSP1-2.6 scenario over mid-latitudes in the Northern Hemisphere, but the model
with the VBS SOA predicted an increase. As a result, VBS SOA over the eastern US was lower than
IEPOX-SOA in present conditions but higher in the 2090s under the SSP1-2.6 scenario, as shown in
Fig. 12. This was because the explicit chemistry took into account aerosol surface area and pH that were
less favorable for IEPOX-SOA formation under the SSP1-2.6 scenario, but the VBS does not consider
aerosol properties in SOA formation (other than OA mass).

VBS SOA decreased over central Africa under the SSP3-7.0 scenario (Fig. 11h), which cannot be
seen in the explicit chemistry (Fig. 5g). The SSP3-7.0 was the scenario with the highest $NO_x$ emissions
among all SSP scenarios (Turnock et al., 2020) and $NO_x$ concentration was predicted to increase over
central Africa (Fig. S8). The VBS SOA yields are lower in the high $NO_x$ pathway than in the low $NO_x$
pathway (Hodzic et al., 2016), which resulted in the VBS SOA decrease over Africa. On the other hand,



the model with explicit chemistry predicted a slight increase of IEPOX-SOA over central Africa under the SSP3-7.0. The $NO_x$ dependency considered in the VBS also applied to the explicit chemistry, but again, the explicit chemistry has other dependencies such as aerosol surface area and pH. The increase of sulfate aerosol concentration over Africa (Fig. S10) promoted IEPOX reactive uptake in the explicit chemistry, while the VBS SOA did not have this dependency.

Figure 12 indicates that isoprene SOA was exceedingly low compared to IEPOX-SOA except for source regions. This result was consistent with Jo et al. (2019), who used the GEOS-Chem chemical transport model (see Fig. 3c in their paper). The main reason was that IEPOX-SOA was nonvolatile but the VBS simulated most of the semi-volatile products as gas phase rather than aerosol phase especially for remote regions (Jo et al., 2019). As a result, global tropospheric burdens of isoprene SOA were lower than those of IEPOX-SOA (e.g., 252 and 101 Gg for IEPOX-SOA and isoprene SOA in present conditions, respectively).

Figure 13 represents the global tropospheric burdens of IEPOX-SOA and isoprene SOA across different model configurations and future climatic scenarios. The inclusion of $CO_2$ inhibition greatly reduced the IEPOX-SOA burden under high forcing scenarios - 1%, 11%, 27%, 41%, and 58% reductions for the present, SSP1-2.6, SSP2-4.5, SSP3-7.0, and SSP5-8.5 scenarios, respectively. The VBS SOA also showed a similar tendency as it followed isoprene emissions - 2%, 8%, 27%, 39%, and 55% reductions for the present, SSP1-2.6, SSP2-4.5, SSP3-7.0, and SSP5-8.5 scenarios, respectively. However, the VBS SOA did not have additional reductions by aerosol pH changes (16–33% reductions), due to a lack of aerosol pH dependency.

The isoprene emission was the main driving factor of IEPOX-SOA burden differences between the SSP scenarios, but Fig. 13 (right column) also showed that chemistry was important, which can change the SOA burden per unit isoprene emission up to 50% (SSP3-7.0 vs SSP1-2.6). However, the values were nearly constant when it comes to the VBS. This suggests that the detailed chemistry should be included for a more accurate SOA sensitivity in models.





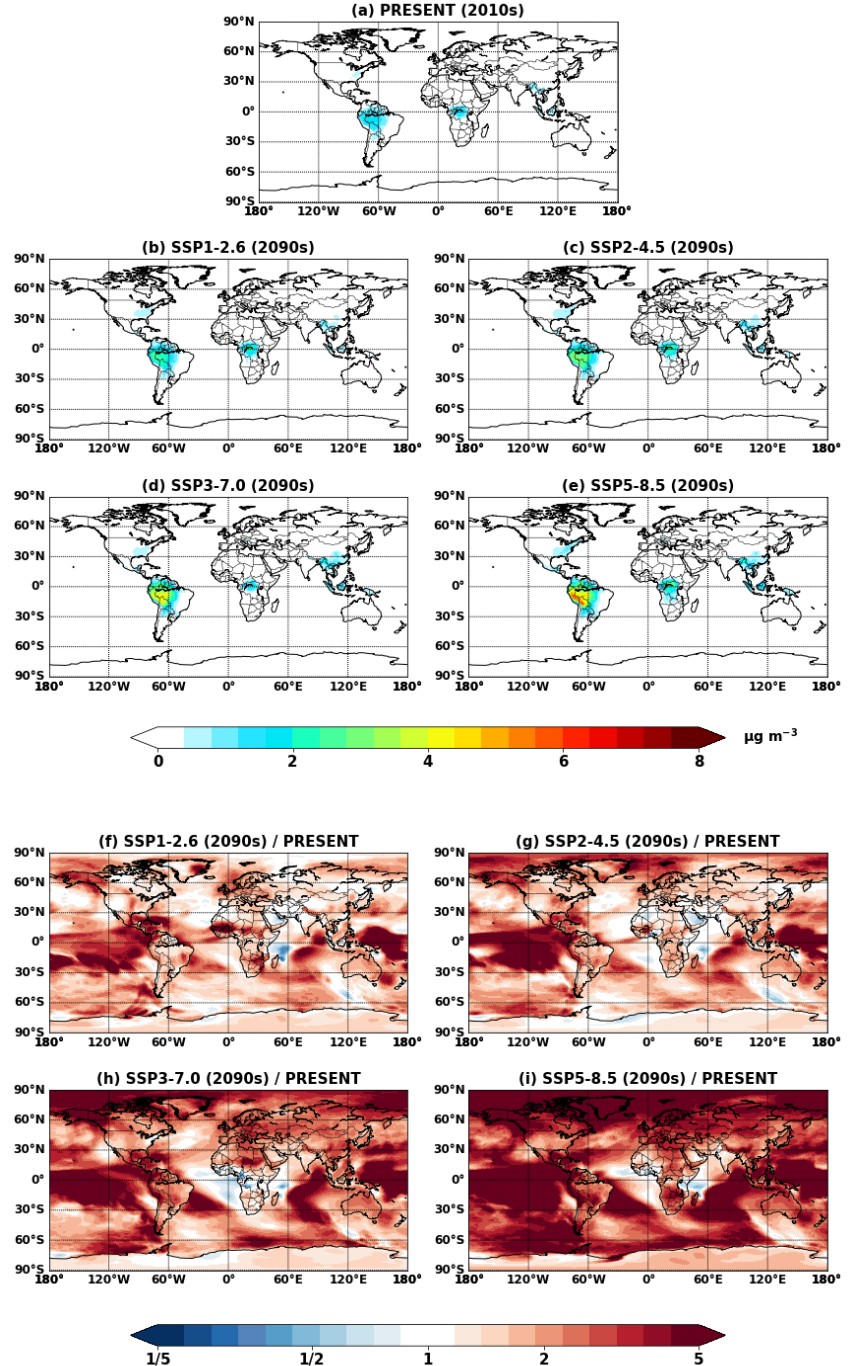

*Figure 11. Global mean isoprene SOA concentrations at the surface simulated in (a) VBS_2010 and*
645 *(b-e) VBS_2090_SSPx. (f-i) Ratios of isoprene SOA between future (b-e) and present (a) conditions.*



*Figure 12. Global ratio maps of surface isoprene SOA by the VBS (VBS) to IEPOX-SOA by the explicit chemistry (EXP) for (a) present (2010s) and (b-e) future (2090s) conditions.*





*Figure 13. Global tropospheric SOA burden (left column) and SOA burden divided by annual isoprene emission (right column). IEPOX-SOA by the explicit chemistry and isoprene SOA by the VBS are shown by filled and hatched bars, respectively. Values in the future are for the 2090s. (a-b) EXP and VBS. (c-d) EXP_CO2 and VBS_CO2 (e-f) EXP_SS_CO2. The VBS SOA is not included in (e-f) as we did not have a simulation for the corresponding case (sea salt included in aerosol pH calculation and $CO_2$ inhibition effect is considered).*

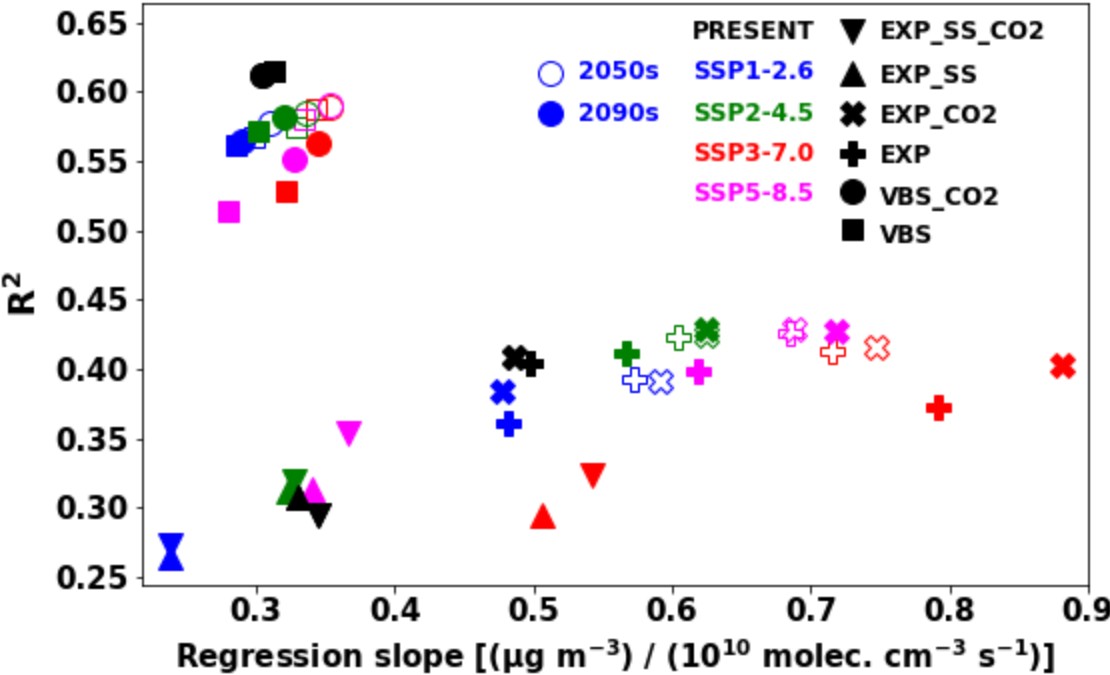

*Figure 14.  Regression slope and R² values between IEPOX-SOA (or VBS SOA) concentrations at the surface and isoprene emissions. Monthly mean values at each model grid point were used for the calculation. The different symbols indicate the different model configurations with filled symbols for the 2090s and open symbols for the 2050s. The colors represent the SSP scenarios (or present conditions) as shown in the legend.*





We further investigated the SOA sensitivities against emissions in different model configurations, as depicted in Fig. 14. For each model configuration in Table 1, we calculated the regression slope and $R^2$ values between monthly gridded IEPOX-SOA (isoprene SOA) concentrations at the surface and isoprene emissions. Similar to the metric (burden/emission) in Fig. 13, the VBS showed a very narrow range of regression slopes. $R^2$ values of the VBS cases were higher than those of the explicit chemistry, which means isoprene SOA generally followed isoprene emissions with little non-linearity. Lower $R^2$ implies other dependencies in simulating SOA concentrations in addition to isoprene emissions. Regression slopes of the models with sea salt in the aerosol pH calculation were lower, because of less acidic conditions. $CO_2$ inhibition also affected the regression slopes under the SSP3-7.0 and SSP5-8.5 scenarios. Higher isoprene emissions without $CO_2$ inhibition made longer lifetimes of IEPOX-SOA precursors (Sect 4.3) and resulted in more IEPOX-SOA in downwind regions. As a result, the regression slopes without $CO_2$ inhibition were lower than with $CO_2$ inhibition, due to less IEPOX-SOA concentrations above isoprene emission source regions. These effects were not apparent in the VBS models.

## 6 Conclusions

Current climate models rely on parameterizations to predict SOA, due to complex non-linear chemistry affecting SOA formation. Thanks to recent progress in the mechanistic understanding of SOA formation from the combined laboratory, field, and modeling studies, chemistry models have started to include explicit SOA mechanisms for a few SOA species including IEPOX-SOA. Here, we implemented a detailed chemical mechanism to explicitly simulate IEPOX-SOA in CESM2.1.0. We investigated future IEPOX-SOA changes under four Tier 1 SSP scenarios, using the explicit chemical mechanism as well as the VBS parameterization.

IEPOX-SOA concentrations were generally predicted to be increased under the SSP scenarios. Isoprene emission change was the primary factor driving IEPOX-SOA change. OH consumption by enhanced isoprene emissions increased lifetimes of IEPOX-SOA precursors, resulting in more



IEPOX-SOA in downwind regions. The $CO_2$ inhibition effect on isoprene emissions became highly influential for high forcing SSP scenarios (SSP3-7.0 and SSP5-8.5); more studies are needed to reduce the high uncertainty related to this effect in future conditions. Aerosol properties (surface area and pH) were an important factor affecting IEPOX-SOA. IEPOX-SOA concentrations were decreased over the Northern Hemisphere mid-latitudes especially under the SSP1-2.6. This was caused by the reduced sulfate aerosols compared to present conditions, and in turn the reduced aerosol surface area and acidity.

We further predicted isoprene SOA concentrations using the VBS scheme. The VBS scheme also simulated the overall increase of isoprene SOA in the future, mainly due to increased isoprene emissions. VBS SOA concentrations were decreased over central Africa under the SSP3-7.0. Increased $NO_x$ concentrations in this region reduced SOA yield, because SOA yields from the high-$NO_x$ pathway are lower than from the low-$NO_x$ pathway. A $NO_x$ dependency was also included in the explicit chemistry mechanism. However, the explicit chemistry further included other dependencies such as aerosol acidity, leading to the opposite trend – the explicit chemistry predicted the slight increase of IEPOX-SOA over central Africa. In terms of the global SOA burden to isoprene emission ratio, the explicit chemistry simulated 50% differences across different SSP scenarios, but the VBS scheme simulated nearly constant values.

The SOA parameterization involves assumptions, inevitably leaving out some chemistry. This can be negligible in simulating present conditions, but can become important under future climate with different emissions and meteorological conditions. Climate models have been improved in terms of resolution, thanks to the recent growth in computational power and parallel computing methods. In addition to the higher spatial and temporal resolution of climate models, the detailed explicit chemistry is also essential for the correct simulation of aerosols, and in turn, radiation and clouds in climate models. With the new development of measurements of the SOA formation mechanism from other compounds in the laboratory, the detailed explicit chemistry of SOA will be available. It would be recommended for climate models to use more comprehensive chemistry, or parameterizations with the



correct physico-chemical dependencies, to simulate more correctly the sensitivity of SOA in future climatic conditions, which often dominates submicron particulate matter mass.


*Code and Data Availability.* CESM2.1.0 is publicly available at https://github.com/ESCOMP/CESM (last access: 2 June 2020). The code updates (isoprene gas-phase mechanism, MOSAIC module, IEPOX-SOA, and NOx-dependent yield of VBS SOA) in this work will be made available as a new compset in future releases of CESM (CESM2.2 and CESM2.3). SEAC[4]RS measurements are available

at https://www-air.larc.nasa.gov/missions/seac4rs/ (last access: 2 June 2020). GoAmazon measurements are available on the Atmospheric Radiation Measurement (ARM) data website: https://www.arm.gov/data (last access: 2 June 2020). ACRIDICON-CHUVA measurements are available on the HALO database: https://halo-db.pa.op.dlr.de/ (last access: 2 June 2020). Surface IEPOX-SOA measurements are available in the supplement of this paper. Model simulation results are

available on the NCAR Digital Asset Service Hub (DASH) at *TBD*.

*Author contributions.* DSJ, AH, LKE, and JLJ designed and performed the research. DSJ performed all model simulations with help from AH, LKE, ST, and MM. RHS updated the isoprene gas-phase chemistry in CESM. PCJ and JLJ analyzed the measurements. RAZ, RCE, BS, and ZL implemented the

MOSAIC module in CESM. WH, CS, JS, JES, AW, PCJ, and JLJ provided observations used in the evaluation of the model. All authors contributed to the review and editing of the paper.

*Competing interests.* The authors declare that they have no conflict of interest.

*Acknowledgments.* This research was supported by grants EPA STAR 83587701-0, DOE (BER, ASR program) DE-SC0016559, NASA 80NSSC18K0630, and NSF AGS-1822664. DSJ was partially supported by an NCAR Advanced Study Program postdoctoral fellowship. AH was supported by the



DOE ASR grant DE-SC0016331. BS, JES, RAZ, and RCE were supported by the U.S. Department of Energy, office of Biological and Environmental Research as part of the Atmospheric System Research
(ASR) and Energy Exascale Earth System Model (E3SM) programs, and used data from the Atmospheric Radiation Measurement (ARM) user facility, a DOE Office of Science User Facility. The Pacific Northwest National Laboratory (PNNL) is operated for DOE by Battelle Memorial Institute under Contract DE-AC05-76RLO1830. Isoprene measurements during SEAC[4]RS were supported by the Austrian Federal Ministry for Transport, Innovation and Technology (bmvit) through the Austrian
Space Applications Programme (ASAP) of the Austrian Research Promotion Agency (FFG). The ACRIDICON-CHUVA aircraft measurements were supported by the Max Planck Society, the German Science Foundation DFG (HALO-SPP 1294, SCHN1138/1-2), and the German Ministry of Research BMBF, grant no. 01LG1205E (ROMIC-SPITFIRE). We thank Jason M. St. Clair, John D. Crounse, and Paul O. Wennberg for ISOPOOH and IEPOX measurements during SEAC[4]RS. Data from the Caltech
CIMS used in this analysis were made possible with support from NASA - NNX12AC06G. We acknowledge high-performance computing support from Cheyenne (doi:10.5065/D6RX99HX) provided by NCAR's Computational and Information Systems Laboratory, sponsored by the National Science Foundation. We thank Kelvin Bates and Loretta Mickley of Harvard, Alex Guenther of Univ. of California Irvine, and Eloise Marais of Univ. of Leicester for useful discussions.



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
