# Peer review of "Future changes in isoprene-epoxydiol-derived secondary organic aerosol (IEPOX-SOA) under the shared socioeconomic pathways: the importance of physico-chemical dependency"

_Atmospheric Chemistry and Physics, 2020_

## Referee Comment (RC1) · Anonymous Referee #1 · 27 Jul 2020

The authors have provided an assessment of the development in a global model of new explicit processes in the formation of SOA. Specifically, the authors have focused on IEPOX derived SOA. The authors compared their implementation with currently used parameterized approaches and sensitivity to future changes in climate and emissions. They found that IEPOX-SOA was predicted to increase in all future scenarios. This was attributed to a combination of isoprene emission changes and acidity influences on reaction chemistry. The new implementation was more responsive to future changes in emissions than the parameterized approaches. These parameterized approaches

predicted nearly constant SOA yield from isoprene emissions regardless of region or type of future scenario. Minor comments below

Section 3 - while the comparison with the observational data was useful I think some intermodel comparison could also provide relevant insights on model behavior. Regardless of the observational locations, where and when were the largest differences in predictions from the parameterized base case to the new model? How did the model do in high urban loading versus more rural locations. Did the changes occur where expected, any surprising predictions?

Line 130 and 151 introduces the scenarios used in the study (SSP5-8.5, SSP3-7.0, SSP2-4.5, and SSP1-2.6). It was hard to follow what unique climatic inputs for each scenario the results were based on in section 4 . Referring to a table could help in briefly listing/defining the differences between scenarios (emissions, anthropogenic activity, etc....) and treatment of the uncertainties within the scenarios either in the main text or in the supplement may help.Ăă

Line 328 - There was no discussion of H sensitivity in section 2.2.2. Could a justification be provided here for the range of H

Line 367 - some context on the implications on the focus on the background. How important are these processes in the background relative to the urban plumes. Would some of the new things that have been instrumented in the model, ie. coatings, may have more impact in urban plumes versus background. Could this be explored in the inter-model comparison?

Line 407 if this is a known experimental bias why can't it be corrected prior to model evaluation?

Line 420 - could more detail be provided on how did the model improve by using the results from D'Ambro et. al?

Line 424 - detailed molecular-level scheme seems to be an important finding here. Can

text be provided with some guidance as to the type or class of molecules need to be considered or are most relevant for future study?

Line 545 - can the changes in Ph be quantified as well as a discussion on the spatial changes in Ph.

Figure 2 c missing quartiles upper altitude?

Figure 11 - can text be provided discussing reasons for driving the predicted changes in South Asia?
* * *

---

## Referee Comment (RC2) · Anonymous Referee #2 · 11 Sep 2020

The authors present a suite of simulations of IEPOX SOA formation using a recent parameterization that they developed (Jo et al. 2019) which uses, in their words, "explicit chemistry".

The idea that VBS or other volatility-based approaches is insufficient to capture the multiphase chemistry IEPOX is not novel, and it is obvious from an atmospheric chemistry and physics point of view. This intercomparison should not be given emphasis in an ACP paper. The community has accepted the idea that multiphase pathways of SOA formation must be represented separately from volatility-based representations

for fifteen years or more.

That being said, the new explicit IEPOX SOA scheme, which has already been peer reviewed and published in Jo et al. 2019 and therefore will not be the focus of my review, is of value, and this manuscript validates it as compared to field data and demonstrates its application to climate predictions as implemented in CESM. The paper would benefit greatly from deeper discussion comparing the performance of this model to other IEPOX SOA chemistry representations in, e.g., CMAQ, GEOS-Chem, WRF-Chem, and the CESM work of the Penner group, rather than the current focus on VBS.

—————————————————————

---

## Author Response (AR1)

**Response to reviewers for the paper "Future changes in isoprene-epoxydiol-derived secondary organic aerosol (IEPOX-SOA) under the shared socioeconomic pathways: the importance of explicit chemistry"**

5 We thank the editor and reviewers for their comments on our paper. To guide the review process we have copied the reviewer comments in black text. Our responses are in regular blue font. We have responded to all the referee comments and made alterations to our paper (**in bold text**). Line numbers are based on the submitted manuscript.

**Editor**

**E1.1)** The Role of Interactions between Inorganic and Organic Aerosol Constituents Altering Aerosol Physicochemical Properties - Self-Limiting Effects of IEPOX SOA Growth:

Zhang et al. (2019, ACS Earth and Space Chem) constructed a core-shell viscosity and thermodynamic model to estimate the values and trends of aerosol viscosity and pH as IEPOX heterogeneously reacted with pure ammonium bisulfate particles during the course of the Riva et al. (2019, ES&T) experiments. The modeling results indicated that at low IEPOX-to-inorganic

20 sulfate ratios (typical of the SE USA), self-limiting reactions of IEPOX-derived SOA are mainly caused by changes in the aerosol phase state; however, as IEPOX concentration increases and thus the IEPOX-to-sulfate ratio increases (more like conditions in the Amazon), the acidity of the aerosol core can influence gamma(IEPOX) as the aerosol pH undergoes more drastic changes initially, followed by the phase state changes due to the formation of the 2-methyltetrol sulfate

25 monomers and then their corresponding OS oligomers. The reason I point this out is that as the authors already know commonly utilized thermodynamic models assume sulfate as only inorganic sulfur; hence, exclusion of particulate OSs may lead to issues of aerosol pH estimations. A challenge in this is that pka values for IEPOX-derived OSs remain uncertain, but Zhang et al. (2019, ACS Earth and Space Chem) explored this through sensitivity analyses. I

30 think the work of Zhang et al. (2019, ACS Earth and Space Chemistry) is worth pointing out here to the authors since it aimed to model the recent experiments by Riva et al. (2019, ES&T). Several of your coauthors on the present study were coauthors on the experimental and field study by Riva et al. (2019, ES&T).

35 Another recent study to note is that Dombeck et al. (2020, J. Environ. Quality) estimated the water-soluble organosulfur contributions to PM2.5 from the IMPROVE Network in 2016. Her work showed that in the eastern U.S. during July 2016 that organosulfur compounds (likely OSs) contribute between ~0.2 - 0.3 ug S m-3. If we assume an average molecular weight of the organosulfates (OSs) to be the IEPOX-derived OS monomer, which is the 2-methyltetrol sulfate

40 diastereomers (MW 216 g mol), then would would estimate that the IEPOX-derived OSs could

contribute 1.35 - 2.03 ug m-3 of total OA to the PM2.5. This is pretty substantial and appears to agree with field estimates from Riva et al. (2019, ES&T). With these types of organosulfate concentrations in ambient PM2.5, one has to wonder how this affects the aerosol phase and pH estimates?

We appreciate the comment that the self-limiting effect can be important especially for IEPOX-SOA over the Amazon region where the IEPOX-to-inorganic sulfate concentration ratio is high. Updating the thermodynamic model to include the impact of organosulfates would be quite complex, and is out of the scope of this study. Instead, we conducted additional sensitivity
50 runs with two assumptions for the thermodynamic model calculation - (1) 90% of inorganic sulfates are converted to organosulfates (upper bound in the atmosphere based on chamber experiments for Amazon condition by Riva et al. (2019)). (2) aerosol pH changes by organics are negligible.

55 Figure E1.1 shows the annual mean pH change for the year 2010 over the Amazon. The map shows aerosol pH changes of 1-2 units over Amazon, which are similar to the results by Zhang et al. (2019).

[Figure]

60 *Figure E1.1. Annual mean aerosol pH at the surface for the year 2010 as simulated by CESM2 model. (a) Base case used in the paper (Base), (b) assuming 90% of inorganic sulfates are converted to organosulfates (Inor90), (c) differences between (a) and (b).*

We evaluated the sensitivity model run (Inor90) against aircraft measurements over Amazon
65 and global surface AMS measurements. Simulated IEPOX-SOA concentrations were substantially decreased as a result of the aerosol pH increase. Model biases changed from positive (75% and 121% for GoAmazon IOP1 and ACRIDICON-CHUVA campaigns, respectively) to negative (-9% and -49%). The model showed improved performance for GoAmazon IOP1 but substantially underestimated the observed IEPOX-SOA concentrations
70 during the ACRIDICON-CHUVA campaign except for below 1 km (Figure E1.2).

[Figure]

*Figure E1.2. Vertical profiles of IEPOX-SOA during GoAmazon IOP1 (a) and ACRIDICON-CHUVA (b) campaigns. Profiles are binned to the 1 km vertical resolution. Boxes show the 25th – 75th percentile with the orange line being the median. Whiskers represent the 10th and 90th percentiles and the black dot indicates the mean value. The sensitivity model results are shown in red lines and the base model results are represented in blue lines (with 50% isoprene emissions from tropical PFTs). The solid line represents mean and the dashed line indicates median values. The number of data points in each km interval is shown at the right of each panel.*

We further compared the modeled IEPOX-SOA against the surface AMS dataset. As expected, assuming 90% of inorganic sulfates are converted to organosulfates resulted in a significant reduction of the global IEPOX-SOA concentration (Figure E1.3). However, if the assumption is applied only for the Amazon region where the relative contribution of organosulfates is high (Figure E1.4), the model showed an improvement in terms of RMSE (0.99 µg m$^{-3}$ to 0.90 µg m$^{-3}$) and $R^2$ (0.62 to 0.83).

[Figure]

*Figure E1.3. Observed global surface IEPOX-SOA concentrations (gray bar) versus the model results assuming 90% of inorganic sulfates are converted to organosulfates globally (solid line). Detailed information about site locations, time period, and IEPOX-SOA concentrations are available in Table S1. The southeast US, the Amazon, and urban regions are shown in blue, green, and red, respectively.*

[Figure]

*Figure E1.4. Observed global surface IEPOX-SOA concentrations (gray bar) versus the model results assuming 90% of inorganic sulfates are converted to organosulfates over the Amazon region only (solid line). Detailed information about site locations, time period, and IEPOX-SOA concentrations are available in Table S2. The southeast US, the Amazon, and urban regions are shown in blue, green, and red, respectively.*

Overall, the sensitivity simulations suggested the potential importance of explicit simulation of organosulfates in aerosol pH calculations, especially over regions where the contribution of organic aerosol is higher. Thermodynamic models used in chemistry models will need to include the explicit treatment of organic aerosols. Based on these results, we added text and figures to the manuscript as follows:

Line 426:

115 **"Unlike volatilities of IEPOX-SOA, even if molecular compositions of IEPOX-SOA are similar for different locations, formation rates of IEPOX-SOA could be different. Depending on the IEPOX-to-inorganic sulfate ratio, aerosol pH can change substantially due to the conversion of inorganic sulfate to organic sulfate (Riva et al., 2019). This inorganic to organic conversion can be especially important for IEPOX-SOA over the**
120 **Amazon, where the conversion was observed to be up to ~90% in the laboratory experiment mimicking the Amazon (Riva et al., 2019). To explore this effect, we conducted a sensitivity simulation, where we assumed that 90% of the inorganic sulfate had converted to organic sulfate, and excluded organic sulfate from the thermodynamic calculation (Sect. 2.2.2). The result is shown in Fig. S7. Model biases were substantially**
125 **reduced, and changed from positive (225% and 394% for GoAmazon IOP1 and ACRIDICON-CHUVA campaigns, respectively) to negative (-9% and -49%). The model showed improved performance for GoAmazon IOP1 but substantially underestimated the observed IEPOX-SOA concentrations during the ACRIDICON-CHUVA campaign except for below 1 km. This change is due to less acidic conditions as shown in Fig. S8, which**
130 **shows aerosol pH increases of 1–2 units over the Amazon, which are similar to the results by Zhang et al. (2019). Given the fact that the IEPOX-to-inorganic sulfate ratio is especially high over Amazon, the modeling of the conversion may improve the model performance over the Amazon while maintaining the model performance over the SE US and other regions. However, it did not improve the comparison especially for the free**
135 **troposphere. The results of this sensitivity run will be further discussed in Sect. 3.3. with other sensitivity model runs."**

Line 444:

140 **"A sensitivity model run with 90% of inorganic sulfate conversion to organic sulfate over the Amazon was also evaluated (Black line in Fig. S10). The overestimation of the model over the Amazon was significantly reduced and became comparable to observed IEPOX-SOA concentrations. $R^2$ was also improved to 0.83 from 0.62."**

145 Line 455:

**"(ii) In terms of model evaluations, the inclusion of organics in aerosol thermodynamic calculations (conversion of inorganic to organic sulfates) seems to be most important, as it improves the model performance in regions where the model overpredicts IEPOX-SOA**
150 **(e.g., the Amazon). This conversion also affects the viscosity of organic layers and increases inhibition of IEPOX reactive uptake rate, because IEPOX-derived organosulfates have likely higher viscosity values than alpha-pinene SOA (Riva et al.,**

2019; Zhang et al., 2019). Although the results are significantly improved over the Amazon, this conversion  will reduce IEPOX-SOA concentrations in regions where the model underestimated the IEPOX-SOA concentrations (e.g., the SE US), even though the conversion fraction over the SE US is expected to be a lot lower than that over the Amazon (Riva et al., 2019)."

[Figure]

*Figure S7. Same as Fig. S6 but assumed 90% of inorganic sulfates are converted to organic sulfates.*

[Figure]

*Figure S8. Annual mean aerosol pH at the surface for the year 2010 as simulated by CESM2 model. (a) Base case used in the paper (Base), (b) assuming 90% of inorganic sulfates are converted to organosulfates (Inor90), (c) difference between (a) and (b).*

165

[Figure]

*Figure S10. Same as Fig. 3 but for (a) the sensitivity model run with 90% of inorganic sulfate conversion to organic sulfate over the Amazon (black line), (b) the sensitivity model run with heterogeneous OH oxidation of IEPOX-SOA using the rate constant of 4.0 × 10⁻¹³ cm³ molec.⁻¹ s⁻¹ (blue line), and (c) IEPOX-SOA simulated by the VBS scheme (red line). For the VBS results, SOA from the isoprene + OH pathway was assumed as IEPOX-SOA. We assumed all aged IEPOX-SOA was lost via the fragmentation process for the heterogeneous OH reaction sensitivity run.*

170

175 **E1.2)** Lines 209-211: The authors are correct that D'Ambro et al. (2019, ACP) confirmed in the lab that IEPOX SOA had a low-volatility character, as originally measured/determined from the field by Hu et al. (2016, ACP) and Lopez-Hilfiker et al. (2016, ES&T). I wanted to note that the development of HILIC/ESI-HR-QTOFMS first by Betsy Stone's group for the detection of multifunctional OSs (Hettiydura et al., 2015, AMT; Hettiydura et al., 2019, ACP) and followed by 180 Cui et al. (2018, ESPI) for measuring IEPOX SOA have helped to demonstrate that most of the IEPOX SOA is likely in the orgnaosulfate form with some contribution of 2-methyltetrols (as

shown by Cui et al., 2018). Cui et al. (2018, ESPI) did show that GC/MS analyses with prior derivatization yielded higher concentrations of 2-methyltetrols compared to HILIC/ESI-HR-QTOFMS when using both standards and lab-generated IEPOX SOA. Further
185  analyses of 2-methyltetrol sulfate standards and IEPOX SOA revealed that C5-alkene triols were produced during GC/MS analyses, suggesting that thermal methods may explain the large productions of C5H12O4 (typically thought to be 2-methyltetrols) or C5H10O3 (typically thought to be C5-alkene triols) signals in the semivolatile ranges measured by FIGAERO-CIMS (Lopez-Hilfiker et al. 2016). Notably, these same ions can be seen in higher temperature
190  thermograms, indicating that there is thermal decomposition that yields these ions detected by I-CIMS.

Thanks for the constructive comment. We added more discussions on IEPOX-SOA molecular tracers, organosulfate, and thermal decomposition as follows:

195

Lines 206-220:

"The SOA yield of IEPOX reactive uptake was assumed to be 100% and IEPOX-SOA was treated as non-volatile in the model. This is consistent with other previous modeling studies
200  (Marais et al., 2016; Budisulistiorini et al., 2017; Stadtler et al., 2018; Schmedding et al., 2019b), based on field studies which showed that ambient IEPOX-SOA has very low volatility (Hu et al., 2016; Lopez-Hilfiker et al., 2016; Riva et al., 2019). A recent chamber-based study by D'Ambro et al. (2019) confirmed the low-volatility of IEPOX-SOA, and suggested that the semivolatile products (**2-methyltetrols, C5-alkene triols, and 3-MeTHF-3,4-diols**) measured by some
205  techniques **using thermal desorption** in ambient IEPOX-SOA mostly resulted from thermal decomposition in those methods. On the other hand, they reported that the semi-volatile 2-methyltetrols that are also formed can evaporate after IEPOX reactive uptake, and can be lost to gas-phase reactions with OH and dry/wet deposition, resulting in an IEPOX-SOA yield lower than unity. However, the evaporation is completed within one hour, and thus is not inconsistent
210  with the very low volatility characteristics of ambient IEPOX-SOA. **The observed volatility of ambient IEPOX-SOA can be explained by the low volatility of organosulfates and oxidation products, which comprise more than half of the IEPOX-SOA tracers (Cui et al., 2018; Chen et al., 2020)**. D'Ambro et al. (2019) also pointed out that the measured $\gamma_{IEPOX}$ is an order of magnitude or more higher than often used in models. We conducted sensitivity tests to
215  investigate the effect of these uncertain parameters on model results in Sect. 3.
**In addition to determining volatility, the formation of organosulfates is also important in terms of aerosol pH, and its effect on the IEPOX-SOA formation rate. Substantial amounts of inorganic sulfate can be converted to organosulfates, especially in regions with a high IEPOX to inorganic sulfate concentration ratio (Riva et al., 2019). Riva et al.**
220  **(2019) showed that up to 90% of inorganic sulfates were converted to organosulfates under laboratory conditions that mimic the Amazon. As a result, the aerosol became less acidic and reduced the reactive uptake of IEPOX (Zhang et al., 2019). Sensitivity simulation was carried out to estimate the changes in IEPOX-SOA concentrations due to**

**inorganic sulfate to organosulfate conversion (Sect. 3). In this sensitivity run, we**
**assumed a 90% conversion as an upper limit in the atmosphere. Because**
**thermodynamic models used in 3D chemistry models (e.g., ISORROPIA, Fountoukis and**
**Nenes (2007)) do not take into account organics (e.g., all models participated in the**
**recent AeroCom phase III, Bian et al. (2017)), we assumed organics have a negligible**
**effect on aerosol pH. Therefore, aerosol pH changes were solely calculated by the loss of**
**inorganic sulfates. There are also high uncertainties in acid dissociation constant (pK$_a$)**
**values for IEPOX-derived organosulfates, which makes it difficult to include**
**organosulfates in the thermodynamic calculation (Zhang et al., 2019), which requires**
**further study."**

**E1.3)** Since work by Riva et al. (2019, ES&T) showed that IEPOX-derived OSs phase separate in inorganic sulfate aerosol to the surface, which agrees with recent thermodynamic predictions (Hyttinen et al., 2020, ACP), Chen et al. (2020, ES&T Lett) recently systematically examined the heterogeneous OH oxidation of particulate 2-methyltetrol sulfates. Notably, they found the chemical lifetime against OH oxidation agreed well with Hu et al. (2016, ACP) and Lam et al. (2019, ACP), but by using HILIC/ESI-HR-QTOFMS, they found that many of the more oxidized and multifunctional OSs previously thought to come from isoprene (as measured in prior lab and field studies) could be produced through heterogeneous OH oxidation of 2-methyltetrol sulfates. A lot of questions remain in how this will affect aerosol phase, morphology, hygroscopicity and acidity. Prior studies using single-particle MS methods, such as the Prather ATOFMS (Hatch et al., 2011, ES&T) and NOAA PALMS (Froyd et al., 2010; Liao et al, 2015), have measured these multifunctional particulate OSs recently demonstrated to form from heterogeneous OH oxidation of the 2-methyltetrol sulfates (Chen et al., 2020, ES&T Letters). In addition, 2-methyltetrol sulfates and these heterogeneously OH aged OS products have been recently measured in cloud water (Pratt et al., 2013, Atmos. Environ.), hailstones (Spolnik et al., 2020, Chemosphere), snow (Spolnik et al., 2020, Chemopshere), and rainwater (Spolnik et al., 2020, Chemosphere) Since 2-methyltetrol sulfates are by far one of the most abundant IEPOX-SOA constituents and likely phase separate (or salt out) to the surface of mixed inorganic-organic particles, this new work really raises the question how will these particles age in a warming and changing atmosphere. I realize the authors may not want to consider aging mechanisms in their future predictions, but maybe this is something worth mentioning as future work? The reason I wanted to point out some of these other OSs that have been observed from isoprene oxidation for a LONG time (e.g., Surratt et al., 2007, ES&T), and until recently we were very unclear how they formed. Some groups proposed that maybe aldehydes could explain some of these particulate OSs, possibly through sulfate radical reactions. But the heterogeneous OH oxidation mechanism (and possibly other aqueous phase processes), may explain these OSs that have been measured offline by LC/ESI-MS methods and by real-time single-particle MS methods. I thought of this since Figure 6 showed no potential chemical loss of IEPOX-SOA.

265  We conducted an additional sensitivity run to explore the global impacts of heterogeneous OH oxidation of IEPOX-SOA. We added a heterogeneous IEPOX-SOA aging reaction in the model with an OH reaction rate constant of $4.0 \times 10^{-13}$ cm$^3$ molec.$^{-1}$ s$^{-1}$. As shown in Figure E1.5, aged IEPOX-SOA is relatively small compared to fresh IEPOX-SOA except in remote oceans far from isoprene sources. This is due to the fact that the lifetime of IEPOX-SOA against heterogeneous

270  oxidation with OH is ~19 days, assuming an average ambient OH concentration of $1.5 \times 10^6$ molec. cm$^{-3}$ (Hu et al., 2016). Chen et al. (2020) also reported that the lifetime of 2-methyltetrol sulfates against heterogeneous OH oxidation was 16 days for OH condition of $1.5 \times 10^6$ molec. cm$^{-3}$, 39 days in the Southeast US and 79 days in Amazon. As shown in Figure 6, the simulated lifetime of IEPOX-SOA was calculated to be ~6 days against wet and dry deposition. Therefore,

275  the heterogeneous aging of IEPOX-SOA is relatively slow and not important except in the remote atmosphere with sufficient aging time.

[Figure]

*Figure E1.5. Simulated annual mean IEPOX-SOA concentrations at the surface. (a) Fresh*
280  *IEPOX-SOA (b) Aged IEPOX-SOA (heterogeneous oxidation against OH). The ratios between fresh and aged IEPOX-SOA are presented in panel (c).*

We added the text as follows:

285  Line 220:

**"Once IEPOX-SOA is formed, there is no further oxidation in the model, as in previous studies (Marais et al., 2016; Budisulistiorini et al., 2017; Schmedding et al., 2019). However, measurement studies have observed further heterogeneous OH oxidation of**
290  **3-methyltetrol sulfate ester (Lam et al., 2019), 2-methyltetrol sulfates (Chen et al., 2020), and IEPOX-SOA factor by positive matrix factorization (PMF) (Hu et al., 2016). Aged IEPOX-SOA can be lost to the gas phase via volatilization if fragmentation is dominant (Hu et al., 2016; Lam et al., 2019), or transformed to highly oxidized molecules if functionalization is more favorable (Chen et al., 2020). Reaction rate constants with OH**
295  **were similar among these studies, $2.0–5.5 \times 10^{-13}$ cm$^3$ molec.$^{-1}$ s$^{-1}$ including error ranges (Hu et al., 2016; Lam et al., 2019; Chen et al., 2020). We used $4.0 \times 10^{-13}$ cm$^3$ molec.$^{-1}$ s$^{-1}$ which is based on the best estimate of Hu et al. (2016), because the model simulates bulk**

**IEPOX-SOA. The sensitivity results of heterogeneous OH oxidation are also discussed in Sect. 3."**

300

Line 444:

**"Another sensitivity model run with heterogeneous OH oxidation of IEPOX-SOA was also investigated (blue line in Fig. S10). Contrary to the inorganic sulfate conversion**
305 **sensitivity run, heterogeneous OH oxidation did not change the model results significantly, even though we assumed 100% loss (via fragmentation) of aged IEPOX-SOA. $R^2$ remained similar (0.62 to 0.61) as simulated IEPOX-SOA concentration changed only slightly. As shown in Fig. S11, Fresh IEPOX-SOA is dominant globally except for over the remote ocean. This is because the IEPOX-SOA lifetime against**
310 **heterogeneous OH oxidation is about ~19 days, assuming an average ambient OH concentration of 1.5 x $10^6$ molec. $cm^{-3}$ (Hu et al., 2016), which is substantially longer than the IEPOX-SOA lifetime against wet and dry deposition (~6 days, Sect. 4)."**

Line 457:

315

**"(iii) The heterogeneous OH reaction was found to be unimportant except over remote regions due to a longer lifetime compared to other loss pathways of IEPOX-SOA in the model. However, this reaction can be important in terms of aerosol-cloud interaction by changing properties of cloud condensation or ice nuclei over remote regions. The**
320 **relative contribution of fragmentation/functionalization and detailed reaction mechanisms with molecular structures should be investigated from observational studies (e.g. newly identified organosulfate molecules from Chen et al. 2020) for future IEPOX-SOA models."**

325

[Figure]

*Figure S11. Simulated annual mean IEPOX-SOA concentrations at the surface. (a) Fresh IEPOX-SOA (b) Aged IEPOX-SOA (after heterogeneous oxidation against OH). The ratios between fresh and aged IEPOX-SOA are presented in panel (c). Aged IEPOX-SOA was assumed to be not evaporated and only lost via wet and dry depositions in the model.*

330

**E1.4)** The authors do an excellent job in including several AMS/ACSM field data from many groups and locations. I was wondering if the authors were interested in data from Budisulistiorini et al. (2016, ACP), where 1 year of continuous measurements were done at both Look Rock and downtown Atlanta. Further, there is another 1-year long study by Rattanavaraha et al. (2017, Atmos. Environ.) for the Centerville site. Obviously, you don't have to say yes, but these are of course other data sets that might be worth modeling against if you want them as another comparison.

We added these datasets to Table S2 (changes shown in Red). We also updated the evaluations to include additional datasets in Figure 3. The results are not much different given that the model still underestimated IEPOX-SOA over the southeast US but overestimated IEPOX-SOA over Amazonia. $R^2$ value was slightly reduced from 0.65 to 0.62.

[Figure]

*Figure 3. Global surface IEPOX-SOA observations (gray bar) versus the model results (solid line). Detailed information about site locations, time period, and IEPOX-SOA concentrations are available in Table S2. The southeast US, the Amazon, and urban regions are shown in blue, green, and red, respectively.*

Table S1. Datasets used in Sect. 3.3 and Fig. 4[a]. Ranges or average plus standard deviation of $f_{C_5H_6O}$ (high resolution) and $f_{82}$ (unit mass resolution) in different studies are also included.

| Name of datasets | Time Period | Site locations and descriptions | Campaign name | Ranges or average±std.dev. $f_{C5H6O}$ (‰) | Ranges or average±std.dev. $f_{82}$ (‰) | OA Conc. (ug/m3) | IEPOX-SOA Conc. (ug/m3) | IEPOX-SOA/OA (%) | Latitude | longitude | Ref. | X axis label in Fig. 4 |
|---|---|---|---|---|---|---|---|---|---|---|---|---|
| **Studies strongly-influenced by isoprene emissions under lower NO** | | | | | | | | | | | | |
| SE US forest - CTR site, 2013 SOAS | Jun-Jul, 2013 | Centreville, AL | SOAS | 6.2±2.4 | 7.6±2.2 | 3.8 | 0.64 | 17 | 32.95 | -87.13 | -1 | Centreville 2013 Summer |
| SE US forest - Look Rock site, 2013 SOAS | Jun-Jul, 2013 | Look Rook | SOAS | N/A | N/A | 4.87 | 1.6 | 33 | 35.61 | -83.55 | -2 | N/A[c] |
| SE US forest - Look Rock site, 2013 Spring | Mar-May, 2013 | Look Rock | N/A | N/A | N/A | 3.23 | 1.32 | 41 | 35.61 | -83.55 | -3 | Look Rock, 2013 Spring |
| SE US forest - Look Rock site, 2013 Summer | Jun-Sep, 2013 | Look Rock | N/A | N/A | N/A | 5.32 | 2.13 | 40 | 35.61 | -83.55 | -3 | Look Rock, 2013 Summer |
| SE US forest - Look Rock site, 2013 Fall | Oct-Dec, 2013 | Look Rock | N/A | N/A | N/A | 2.83 | 0.76 | 27 | 35.61 | -83.55 | -3 | Look Rock, 2013 Fall |
| Atlanta JST site, 2012 Spring | Mar-Jun, 2012 | Urban JST site, Atlanta, Georgia, US | N/A | N/A | N/A | 4.7 | 1.74 | 37 | 33.78 | -84.42 | -3 | Atlanta (JST), 2012 Spring |
| Atlanta JST site, 2013 Summer | Jul-Sep, 2013 | Urban JST site, Atlanta, Georgia, US | N/A | N/A | N/A | 6.15 | 2.34 | 38 | 33.78 | -84.42 | -3 | Atlanta (JST), 2013 Summer |
| Atlanta JST site, 2014 Spring | May-Jun, 2014 | Urban JST site, Atlanta, Georgia, US | N/A | N/A | N/A | 9.61 | 2.4 | 25 | 33.78 | -84.42 | -4 | Atlanta (JST), 2014 Spring |
| Atlanta JST site, 2014 Summer | Jul-Sep, 2014 | Urban JST site, Atlanta, Georgia, US | N/A | N/A | N/A | 11.36 | 3.29 | 29 | 33.78 | -84.42 | -4 | Atlanta (JST), 2014 Summer |
| Pristine Amazon forest 2008, Brazil | Feb-Mar, 2008 | Pristine rain forest site, TT34 | AMAZE-08 | 5.0±2.3 | 7.9±1.7 | 0.76 | 0.26 | 34 | -2.59 | -60.2 | -5 | Amazon, 2008 Summer |

| | | | | | | | | | | | | |
|---|---|---|---|---|---|---|---|---|---|---|---|---|
| Amazon forest downwind Manaus, Brazil | Feb-Mar, 2014 | T3 site, near Manacapuru | GoAmazon2014/5 | 6.9±1.6 | 7.1±1.0 | 1.3 | 0.286 | 22 | -3.21 | -60.59 | -6 | Amazon, 2014 Summer |
| Pristine Amazon forest 2014, Brazil | Aug-Dec, 2014 | T0 site, ~150 km northeast of Manaus | GoAmazon2014/5 | N/A | 5.6±1.7 | N/A | N/A | N/A | -3.21 | -60.59 | -7 | N/A |
| SEAC4RS | Aug-Sep, 2013 | Aircraft measurement | SEAC4RS | 4.3±1.6 | N/A | N/A | N/A | 32 | Flight track | Flight track | -8 | N/A |
| Borneo forest, Malaysia | Jun-Jul, 2008 | Rain forest GAW station, Sabah, Malaysia | OP3 | 10±0.3 | 12.4±0.4 | 0.75 | 0.18 | 24 | 4.981 | 117.844 | -9 | Borneo, 2008 Summer |
| Atlanta JST site, 2011 Summer | Aug-Sep, 2011 | Urban JST site, Atlanta, Georgia, US | N/A | N/A | 3.7±1.9 | 11.6 | 3.8 | 33 | 33.78 | -84.42 | -10 | Atlanta (JST), 2011 Summer |
| Atlanta JST site, 2012 May | May, 2012 | Urban JST site, Atlanta, Georgia, US | N/A | 3.3±0.9 | N/A | 9.1 | 1.91 | 21 | 33.78 | -84.42 | -11 | N/A[d] |
| Atlanta GT site, 2012 Summer | Aug, 2012 | Urban Georgia Tech site, Georgia, US | N/A | 5.4±1.9 | N/A | 9.6 | 3 | 31 | 33.78 | -84.396 | -11 | Atlanta (GT), 2012 Summer |
| Yorkville, 2012 Summer | July, 2012 | Rural sites, 80km northwest of JST site, Georgia, US | N/A | 7.7±2.2 | N/A | 11.2 | 4 | 36 | 33.9285 | -85.045 | -11 | Yorkville, 2012 Summer |
| Harrow, Canada | Jun-Jul, 2007 | Harrow site, rural sites surrounded by farmland, Canada | BAQSMET | N/A | N/A | N/A | N/A | 17 | 42.03 | -82.9 | -12 | N/A |
| Bear Creek, Canada | Jun-Jul, 2007 | Bear Creek site, wetlands area surrounded by farmland, Canada | BAQSMET | N/A | N/A | N/A | N/A | 6 | 42.51 | -82.34 | -12 | N/A |
| **Studies strongly-influenced by monoterpene emissions** | | | | | | | | | | | | |
| Rocky mountain pine forest, CO, USA | Jul-Aug, 2011 | Manitou Experimental Forest Observatory, CO, | BEACHON-RoMBAS | 3.7±0.5 | 5.1±0.5 | N/A | N/A | N/A | 39.1 | -105.1 | -13 | N/A |
| European Boreal forest, Finland | 2008-2009 | Hyytiala site in Pine forest, Finland | EUCAARI campaign | 2.5±0.1 [b] | 4.8±0.1 [b] | N/A | N/A | N/A | 61.85 | 24.28 | -9 | N/A |

| | | | | | | | | | | | | |
|---|---|---|---|---|---|---|---|---|---|---|---|---|
| **Studies mixed-influenced by isoprene and monoterpene emissions** | | | | | | | | | | | | |
| North American temperate, US | Aug-Sep, 2007 | Blodgett Forest Ameriflux Site, CA, US | BEARPEX | 4.0±<0.1 [b] | 4.0±<0.1 [b] | N/A | N/A | N/A | N/A | N/A | -9 | N/A |
| **Studies strongly-influenced by urban emissions** | | | | | | | | | | | | |
| Los Angeles area , CA, USA | May-Jun, 2010 | Pasadena, US | CalNex | 1.6±0.2 | 3.6±0.5 | 7 | <DL | < PMF limit | 34.14 | -118.12 | -14 | Pasadena 2010 Spring |
| Beijing, China | Nov-Dec, 2010 | Peking University, in NW of Beijing city, China | N/A | 1.5±0.3 | 4.6±0.7 | 34.5 | <DL | < PMF limit | 39.99 | 116.31 | -15 | Beijing 2010 Winter |
| Changdao island, Downwind of China | Mar-Apr, 2011 | Changdao island, China | CAPTAIN | 1.6±0.2 | 3.8±0.5 | 13.4 | <DL | < PMF limit | 37.99 | 120.7 | -16 | Changdao 2011 Spring |
| Barcelona area, Spain | Feb-Mar, 2009 | Montseny, Spain | DAURE | 1.6±0.2 | 4.8±0.9 | N/A | <DL | < PMF limit | 41.38 | 2.1 | -17 | Montseny 2009 Spring |

a- HR-ToF-AMS was used for all the campaigns except the Atlanta, US, Look Rock, US, and Pristine Amazon forest 2014, Brazil using ACSM.

b- Standard error

c- included in Look Rock 2013 Summer

d- included in Atlanta (JST) 2012 Spring

(1)(Hu et al., 2015b); (2)(Budisulistiorini et al., 2015); (3)(Budisulistiorini et al., 2016); (4)(Rattanavaraha et al., 2017); (5)(Chen et al., 2014); (6)(de Sá et al., 2017); (7)(Carbone et al., 2015); (8)(Liao et al., 2014) ; (9)(Robinson et al., 2011); (10)(Budisulistiorini et al., 2013); (11)(Xu et al., 2014; Xu et al., 2015); (12)(Slowik et al., 2011); (13)(Ortega et al., 2014); (14)(Hayes et al., 2013); (15)(Hu et al., 2015a); (16)(Hu et al., 2013); (17)(Minguillón et al., 2011)

**Anonymous Referee #1**

**R1.0)** The authors have provided an assessment of the development in a global model of new explicit processes in the formation of SOA. Specifically, the authors have focused on IEPOX derived SOA. The authors compared their implementation with currently used parameterized approaches and sensitivity to future changes in climate and emissions. They found that IEPOX-SOA was predicted to increase in all future scenarios. This was attributed to a combination of isoprene emission changes and acidity influences on reaction chemistry. The new implementation was more responsive to future changes in emissions than the parameterized approaches. These parameterized approaches predicted nearly constant SOA yield from isoprene emissions regardless of region or type of future scenario. Minor comments below.

**R1.1)** Section 3 - while the comparison with the observational data was useful I think some intermodel comparison could also provide relevant insights on model behavior. Regardless of the observational locations, where and when were the largest differences in predictions from the parameterized base case to the new model? How did the model do in high urban loading versus more rural locations. Did the changes occur where expected, any surprising predictions?

Thanks for the constructive comment. Comparisons between the VBS and the explicit chemistry are presented in Sect. 5, in terms of global spatial distribution (Figure 12) and burden (Figure 13). Compared to the explicit scheme, the VBS generally predicted less IEPOX-SOA except for regions with high preexisting aerosol levels (Fig. 12a). Therefore, both schemes showed comparable aerosol results in high urban loading conditions (e.g., The eastern US, Asia), but the VBS predicted very low IEPOX-SOA concentrations (e.g., over the ocean). However, even in rural locations, both schemes showed similar IEPOX-SOA concentrations when there were high biogenic SOA loadings (e.g., the Amazon, Borneo). We added the text in Sect. 5 as follows:

Line 627:

**Therefore, both schemes simulated comparable results in high aerosol loading conditions including urban (e.g. the eastern US, Asia) as well as biogenic-dominated regions (e.g., the Amazon, Borneo), while the VBS predicted very low IEPOX-SOA concentrations in most regions with low aerosol levels.**

We further compared these models in terms of global evaluations in Sect. 3. The evaluation of the VBS scheme against global AMS observation showed substantial underestimation over the SE US and overestimation over the Amazon and Borneo (red line in Figure S10). We added Figure S10 and the text as follows:

**"Finally, we evaluated IEPOX-SOA concentrations simulated by the VBS (red line in Fig. S10). The VBS scheme also showed the underestimation of IEPOX-SOA over the SE US and the overestimation over the Amazon. However, the VBS scheme substantially underestimated the observed IEPOX-SOA over the SE US, and $R^2$ value (0.42) was significantly decreased. The explicit scheme showed better performance than the VBS scheme in terms of both bias and variability."**

[Figure]

*Figure S10. Same as Fig. 3 but for (a) the sensitivity model run with 90% of inorganic sulfate conversion to organic sulfate over the Amazon (black line), (b) the sensitivity model run with heterogeneous OH oxidation of IEPOX-SOA using the rate constant of 4.0 × 10$^{-13}$ cm$^3$ molec.$^{-1}$ s$^{-1}$ (blue line), and (c) IEPOX-SOA simulated by the VBS scheme (red line). For the VBS results, SOA from the isoprene + OH pathway was assumed as IEPOX-SOA. We assumed all aged IEPOX-SOA was lost via the fragmentation process for the heterogeneous OH reaction sensitivity run.*

**R1.2)** Line 130 and 151 introduces the scenarios used in the study (SSP5-8.5, SSP3-7.0, SSP2-4.5, and SSP1-2.6). It was hard to follow what unique climatic inputs for each scenario the results were based on in section 4. Referring to a table could help in briefly listing/defining the differences between scenarios (emissions, anthropogenic activity, etc. ...) and treatment of the uncertainties within the scenarios either in the main text or in the supplement may help.

We added the information about different SSP scenarios used in this study in the main text and Table S1 as follows:

 **"A detailed description of each SSP scenario is briefly summarized in Table S1 (narrative, forcing category, population, regulation, and emission amount) and Fig. S1 (emission trajectories for $SO_2$ and $NO_x$)."**

*Table S1. A brief summary of SSP scenarios used in this study. Values are for the year*
 *2100. For more information, readers are referred to previous studies (O'Neill et al., 2016; Riahi et al., 2017; Kc and Lutz, 2017; Gidden et al., 2019; Feng et al., 2020).*

| | | |
|---|---|---|
| SSP1-2.6 | Title[1] | Sustainability - Taking the Green Road (Low challenges to mitigation and adaptation) |
| | Description[1] | The world shifts gradually, but pervasively, toward a more sustainable path, emphasizing more inclusive development that respects perceived environmental boundaries. Management of the global commons slowly improves, educational and health investments accelerate the demographic transition, and the emphasis on economic growth shifts toward a broader emphasis on human well-being. Driven by an increasing commitment to achieving development goals, inequality is reduced both across and within countries. Consumption is oriented toward low material growth and lower resource and energy intensity. |
| | Forcing category[2] | Low |
| | Target forcing level[2] ($W\ m^{-2}$) | 2.6 |
| | Population[3] (millions) | 6,881 |
| | Land use change regulation[1] | strong |
| | Sulfur emissions[4] ($Mt\ SO_2\ yr^{-1}$) | 8.1 |
| | $NO_x$ emissions[4] ($Mt\ NO_2\ yr^{-1}$) | 41.2 |
| | VOC emissions[4] ($Mt\ VOC\ yr^{-1}$) | 62.3 |
| | OC emissions[4] ($Mt\ OC\ yr^{-1}$) | 13.1 |
| | $CO_2$ emissions[4] ($Mt\ CO_2\ yr^{-1}$) | -8,618 |
| SSP2-4.5 | Title[1] | Middle of the Road (Medium challenges to mitigation and adaptation) |

| | | |
|---|---|---|
| | Description[1] | The world follows a path in which social, economic, and technological trends do not shift markedly from historical patterns. Development and income growth proceeds unevenly, with some countries making relatively good progress while others fall short of expectations. Global and national institutions work toward but make slow progress in achieving sustainable development goals. Environmental systems experience degradation, although there are some improvements and overall the intensity of resource and energy use declines. Global population growth is moderate and levels off in the second half of the century. Income inequality persists or improves only slowly and challenges to reducing vulnerability to societal and environmental changes remain. |
| | Forcing category[2] | Medium |
| | Target forcing level[2] (W m$^{-2}$) | 4.5 |
| | Population[3] (millions) | 9,000 |
| | Land use change regulation[1] | medium |
| | Sulfur emissions[4] (Mt $SO_2$ yr$^{-1}$) | 30.8 |
| | $NO_x$ emissions[4] (Mt $NO_2$ yr$^{-1}$) | 77.7 |
| | VOC emissions[4] (Mt VOC yr$^{-1}$) | 120.7 |
| | OC emissions[4] (Mt OC yr$^{-1}$) | 14.5 |
| | $CO_2$ emissions[4] (Mt $CO_2$ yr$^{-1}$) | 9,683 |
| SSP3-7.0 | Title[1] | Regional Rivalry – A Rocky Road (High challenges to mitigation and adaptation) |
| | Description[1] | A resurgent nationalism, concerns about competitiveness and security, and regional conflicts push countries to increasingly focus on domestic or, at most, regional issues. Policies shift over time to become increasingly oriented toward national and regional security issues. Countries focus on achieving energy and food security goals within their own regions at the expense of broader-based development. Investments in education and technological development decline. Economic development is slow, consumption is material-intensive, and inequalities persist or worsen over time. Population growth is low in industrialized and high in developing countries. A low international priority for addressing environmental concerns leads to strong environmental degradation in some regions. |
| | Forcing category[2] | High |

| | | |
|---|---|---|
| | Target forcing level[2] (W m$^{-2}$) | 7.0 |
| | Population[3] (millions) | 12,627 |
| | Land use change regulation[1] | weak |
| | Sulfur emissions[4] (Mt SO$_2$ yr$^{-1}$) | 78.1 |
| | NO$_x$ emissions[4] (Mt NO$_2$ yr$^{-1}$) | 144.4 |
| | VOC emissions[4] (Mt VOC yr$^{-1}$) | 227.9 |
| | OC emissions[4] (Mt OC yr$^{-1}$) | 33.7 |
| | CO$_2$ emissions[4] (Mt CO$_2$ yr$^{-1}$) | 82,726 |
| SSP5-8.5 | Title[1] | Fossil-fueled Development – Taking the Highway (High challenges to mitigation, low challenges to adaptation) |
| | Description[1] | This world places increasing faith in competitive markets, innovation and participatory societies to produce rapid technological progress and development of human capital as the path to sustainable development. Global markets are increasingly integrated. There are also strong investments in health, education, and institutions to enhance human and social capital. At the same time, the push for economic and social development is coupled with the exploitation of abundant fossil fuel resources and the adoption of resource and energy intensive lifestyles around the world. All these factors lead to rapid growth of the global economy, while global population peaks and declines in the 21st century. Local environmental problems like air pollution are successfully managed. There is faith in the ability to effectively manage social and ecological systems, including by geo-engineering if necessary. |
| | Forcing category[2] | High |
| | Target forcing level[2] (W m$^{-2}$) | 8.5 |
| | Population[3] (millions) | 7,363 |
| | Land use change regulation[1] | medium |
| | Sulfur emissions[4] (Mt SO$_2$ yr$^{-1}$) | 29.5 |
| | NO$_x$ emissions[4] (Mt NO$_2$ yr$^{-1}$) | 98.7 |
| | VOC emissions[4] (Mt VOC yr$^{-1}$) | 163.3 |
| | OC emissions[4] (Mt OC yr$^{-1}$) | 17.6 |
| | CO$_2$ emissions[4] (Mt CO$_2$ yr$^{-1}$) | 126,287 |

1) Riahi et al. (2017); 2) O'Neill et al. (2016); 3) KC and Lutz (2017); 4) Gidden et al. (2019)

[Figure]

*Figure S1. Emission trajectories for SO₂ and NOₓ for the four Tier 1 scenarios used in this study.*

**R1.3)** Line 328 - There was no discussion of H sensitivity in section 2.2.2. Could a justification be provided here for the range of H

We provided a range of Henry's law constants of IEPOX used in previous studies and added the text as follows:

Lines 328-330:

**"We conducted an additional sensitivity test as discussed in Sect. 2.2.2. We decreased the IEPOX-SOA yield from IEPOX to 0.2 (based on Fig. 8 of D'Ambro et al. (2019)). To increase $\gamma_{IEPOX}$ in the model, we increased the effective Henry's law constant (H*) which is highly uncertain and spans two orders of magnitude in previous literature (1.0 x 10⁶ ~ 1.7 x 10⁸ M atm⁻¹). Previous studies have used 1.0 x 10⁶ M atm⁻¹ (Zhang et al., 2019), 2.7 x 10⁶ M atm⁻¹ (Pye et al., 2013; Lin et al., 2016), 1.7 x 10⁷ M atm⁻¹ (Zheng et al., 2020), 3.0 x 10⁷ M atm⁻¹ (Budisulistiorini et al., 2017; Pye et al., 2017; Zhang et al., 2018a), 3.3 x 10⁷ M atm⁻¹ (Marais et al., 2016), 1.3 x 10⁸ M atm⁻¹ (Eddingsaas et al., 2010), and 1.7 x 10⁸ M atm⁻¹ (Gaston et al., 2014a). We increased H* by a factor of 5 (8.5 × 10⁷ M atm⁻¹ from 1.7 × 10⁷ M atm⁻¹)."**

**R1.4)** Line 367 - some context on the implications on the focus on the background. How important are these processes in the background relative to the urban plumes. Would some of the new things that have been instrumented in the model, ie. coatings, may have more impact in urban plumes versus background. Could this be explored in the inter-model comparison?

455

Unfortunately, the global model (0.95 in latitude by 1.25 in longitude) is too coarse to investigate the contrast between urban plume and background conditions. To resolve urban plumes, typically regional models with very fine resolution (spatial scales of a few km) are needed (e.g. Shrivastava et al. (2019) used WRF-Chem with 2 km grid spacing to resolve urban plumes over 460 the Amazon).

Currently, the regional refinement version of the CESM2/CAM6-chem model is being developed, which can simulate air pollutants with a regionally refined grid of ~14 km resolution or finer over the region of interest (Pfister et al., 2020). This new capability will enable resolving urban 465 plumes from background conditions. We added the discussion regarding future studies in Sect. 3.3. as follows.

Line 457:

470 **"It is worth noting that there is also a limitation with the coarse grid resolution of the global model (0.95° x 1.25°) in addition to chemistry. The model representativeness may be insufficient in some observation locations. Currently, the regional refinement version of the CESM2/CAM6-chem model is under development, which can simulate air pollutants with a regionally refined grid of ~14 km resolution or finer over the region of 475 interest while maintaining the capability of simulating the whole globe at 1-degree resolution (Pfister et al., 2020). This multiscale model will be able to better represent specific observational sites."**

**R1.5)** Line 407 if this is a known experimental bias why can't it be corrected prior to model 480 evaluation?

We don't have correction factors to adjust the HALO AMS measurements below 2 km. Instead, Figure 11 in Mei et al. (2020) showed the differences of vertical concentration profiles between the two aircraft AMS measurements (G1 and HALO), and the reason behind the discrepancy 485 was thought to be particle losses in the constant pressure inlet used on the HALO AMS. However, the discrepancy could also be partially caused by different air masses sampled in two different aircraft flights, although Mei et al. (2020) used several strict criteria for the comparison. In addition, we would like to keep the measurement data in the figure to avoid misreading or confusion in the interpretation. Instead, we provided another model bias based on increasing 490 the observed values below 2 km by a factor of two in the manuscript as described below.

Line 411:

**"The model biases during the ACRIDICON-CHUVA were reduced to 88-641% after increasing the observed IEPOX-SOA concentrations below 2 km by two times."**

**R1.6)** Line 420 - could more detail be provided on how did the model improve by using the results from D'Ambro et. al?

The model was improved in terms of bias. We noted this in the text as follows:

Lines 418-420:

"However in general, contrary to the evaluation against SEAC4RS measurements (Sect. 3.1), the inclusion of the recent findings from the chamber study by D'Ambro et al. (2019) improved the model simulation **in terms of model bias (225–394% to 11–28%)."**

**R1.7)** Line 424 - detailed molecular-level scheme seems to be an important finding here. Can text be provided with some guidance as to the type or class of molecules need to be considered or are most relevant for future study?

We appreciate the comment that a more detailed discussion about the molecular-level understanding is important for future modeling studies. We also conducted additional sensitivity tests to enrich the discussion (see E1.1 and E1.3).

Line 455:

**"Three sensitivity model runs in this section indicate that the accurate prediction of aerosol composition is key for predicting IEPOX-SOA. (i) If newly formed IEPOX-SOA is mainly composed of semi-volatile species (e.g. 2-methyltetrols) instead of very low volatile species (e.g. organosulfates), it is likely to undergo evaporation, and the effective IEPOX-SOA yield should be decreased in the model unless the model explicitly calculates reevaporation (D'Ambro et al., 2019). (ii) In terms of model evaluations, the inclusion of organics in aerosol thermodynamic calculations (conversion of inorganic to organic sulfates) seems to be most important, as it improves the model performance in regions where the model overpredicts IEPOX-SOA (e.g., the Amazon). This conversion also affects the viscosity of organic layers and increases inhibition of IEPOX reactive uptake rate, because IEPOX-derived organosulfates have likely higher viscosity values than alpha-pinene SOA (Riva et al., 2019; Zhang et al., 2019). Although the results are significantly improved over the Amazon, this conversion will reduce IEPOX-SOA concentrations in regions where the model underestimated the IEPOX-SOA concentrations (e.g., the SE US), even though the conversion fraction over the SE US is**

**expected to be a lot lower than that over the Amazon (Riva et al., 2019). (iii) The heterogeneous OH reaction was found to be unimportant except over remote regions due to a longer lifetime compared to other loss pathways of IEPOX-SOA in the model. However, this reaction can be important in terms of aerosol-cloud interaction by changing properties of cloud condensation or ice nuclei over remote regions. The relative contribution of fragmentation/functionalization and detailed reaction mechanisms with molecular structures should be investigated from observational studies (e.g. newly identified organosulfate molecules from Chen et al. 2020) for future IEPOX-SOA models. Overall, sensitivity studies revealed that future models will need to include a more molecular-level based treatment of SOA. However, there are many uncertain parameters that should be investigated first, such as the molecular composition of fresh/aged IEPOX-SOA and resulting volatility, viscosity, acidity, $pK_a$, hygroscopicity, etc.”**

**R1.8)** Line 545 - can the changes in Ph be quantified as well as a discussion on the spatial changes in Ph.

We added the text and figures for aerosol pH and IEPOX-SOA changes due to the inclusion of sea salt aerosols in the aerosol pH calculations.

Line 549:

**“Most of IEPOX-SOA decreases were over the Amazon (Fig. S17) where absolute IEPOX-SOA concentrations were highest (Fig. 7b-e). Aerosol pH increased mainly over the ocean but also increased over the Amazon due to the transport of sea salt aerosols by trade winds (Fig. S18).** This result indicates that the accurate treatment of inorganic aerosols **and their conversion to organic sulfates,** and related thermodynamic calculations are critically needed for the accurate simulation of organic aerosols.”

[Figure]

**Figure S17. Simulated IEPOX-SOA concentration changes at the surface by including sea salt in aerosol thermodynamic calculation (EXP_SS_2090 - EXP_2090).**

565

[Figure]

*Figure S18. The multi-year mean global aerosol pH (accumulation mode) of EXP_SS_2090 simulations at the surface (left column) and pH differences between EXP_SS_2090 and EXP_2090 simulations.*

570

**R1.9)** Figure 2 c missing quartiles upper altitude?

The median and quartile values were not visible at the 3-4 km level in Figure 3c, because values were below 0.001 µg m$^{-3}$ (75th percentile was 0.0001 µg m$^{-3}$). We added the information in the figure caption.

*"Figure 2. Vertical profiles of isoprene (a,b) and IEPOX-SOA (c,d) during the GoAmazon (a,b,c) and ACRIDICON-CHUVA (d) campaigns. Profiles are binned to the 1 km vertical resolution. Boxes show the 25$^{th}$ – 75$^{th}$ percentile with the orange line showing the median. Whiskers represent the 10th and 90$^{th}$ percentiles and the black dot indicates the mean value. The model results are shown in blue, red, and green for the base, half, and quarter isoprene emissions from Tropical tree PFTs, respectively. The solid line represents mean and the dashed line indicates median values. **The observed median and quartile values are not visible at the 3–4 km level in panel (c), as they are smaller than 0.001 µg m$^{-3}$.** The number of data points in each km interval is shown at the right of each panel."*

**R1.10)** Figure 11 - can text be provided discussing reasons for driving the predicted changes in South Asia?

Isoprene SOA concentrations are very low over the South Asia region (Annual mean concentration at the surface is ~0.05 µg m$^{-3}$ over the South Asia region [60–90°E, 5–40°N], Fig. 11a), so we think it is not necessary to include further discussions about changes in South Asia in the main text. We rather discuss the changes in this response. Isoprene SOA generated by the VBS scheme is mainly affected by isoprene, OH, and preexisting OA concentrations. Figures are attached for each chemical change below. First, OH in South Asia remained similar in all SSP scenarios (Figure R1.1). Isoprene concentrations were increased in all four scenarios, but the magnitudes were different. Increases were highest in SSP1-2.6 and lowest in SSP3-7.0 (Figure R1.2). On the other hand, OA concentrations were predicted to increase in SSP-3.7.0 but substantially decrease in SSP1-2.6 (Figure R1.3). Because both precursor and preexisting aerosol concentrations are proportional to SOA concentration by the VBS, the highest (lowest) isoprene concentrations and the lowest (highest) OA concentrations cancel each other so that the predicted SOA concentration in South Asia is almost unchanged.

[Figure]

605 **Figure R1.1.** *Global ratios of OH concentrations under future SSP scenarios (2090s) to present conditions (2010s) at the surface (VBS case).*

[Figure]

**Figure R1.2.** *Global ratios of isoprene concentrations under future SSP scenarios (2090s) to present conditions (2010s) at the surface (VBS case).*

[Figure]

610

**Figure R1.3.** *Global ratios of OA concentrations under future SSP scenarios (2090s) to present conditions (2010s) at the surface (VBS case).*

**Anonymous Referee #2**

615

**R2.1)** The authors present a suite of simulations of IEPOX SOA formation using a recent parameterization that they developed (Jo et al. 2019) which uses, in their words, "explicit Chemistry".

620 The idea that VBS or other volatility-based approaches is insufficient to capture the multiphase chemistry IEPOX is not novel, and it is obvious from an atmospheric chemistry and physics point of view. This intercomparison should not be given emphasis in an ACP paper. The community has accepted the idea that multiphase pathways of SOA formation must be represented separately from volatility-based representations for fifteen years or more.

625 That being said, the new explicit IEPOX SOA scheme, which has already been peer reviewed and published in Jo et al. 2019 and therefore will not be the focus of my review, is of value, and this manuscript validates it as compared to field data and demonstrates its application to climate predictions as implemented in CESM. The paper would benefit greatly from deeper discussion comparing the performance of this model to other IEPOX SOA chemistry representations in,

630 e.g., CMAQ, GEOS-Chem, WRF-Chem, and the CESM work of the Penner group, rather than the current focus on VBS.

To our knowledge, there have been no future climate studies focusing on IEPOX-SOA changes under different future climatic scenarios or comparing the explicit and the volatility-lumped

635 parameterizations. We also provided comprehensive evaluations of the current IEPOX-SOA mechanism, which have been done in previous studies only for much more limited datasets with narrow spatial and time coverage (e.g. SOAS and SEAC4RS). We also conducted additional sensitivity runs based on recent findings from laboratory studies and gave suggestions for future research in IEPOX-SOA modeling. We further investigated future IEPOX-SOA changes from

640 various aspects, such as isoprene emission increase and resulting OH decrease due to OH consumption, lifetime increases of precursors and longer transport times in downwind regions, changes of the relative contribution of peroxy radical fates (with NO, $HO_2$, isomerization, and other minor pathways), changes of aerosol formation rate due to different aerosol properties in different future scenarios. Previous studies have only focused on emission, OH, and

645 temperature changes. For example, even the recent previous studies that focused on biogenic SOA changes in the future investigated only biogenic SOA changes due to biogenic VOC emission changes or meteorological field changes (Sporre et al., 2019; Liu et al., 2019). Cholakian et al. (2019) further investigated the effects of temperature on biogenic SOA partitioning with the two product and VBS schemes, but the analysis was still limited without the

650 investigation into detailed chemical pathways or aerosol properties.

We agree that there are previous studies that showed volatility-based parameterizations are insufficient to capture IEPOX-SOA changes in response to emission changes. However, VBS parameterizations continue to be used in many models, and there have been no comprehensive

studies comparing the explicit approach and the VBS in different future scenarios. Furthermore, we showed that even if the VBS predicted different SOA concentrations/burden under different future scenarios, it's mainly caused by isoprene emissions. Because the VBS scheme has more dependencies than the two product or simple yield schemes (e.g., more temperature dependency with several volatility bins, $NO_x/HO_2$ dependency), we could expect most of the SOA predictions under future climate projection could have a wrong response to climate changes. Therefore, we wanted to inform future studies focusing on SOA changes in future climates, by providing quantitative comparisons that have not been reported in previous studies. The message is that although climate models can be computationally expensive and so try to reduce the computational cost, simplifying SOA schemes too much would result in wrong projections of SOA concentrations, cloud, radiative forcing, etc. Detailed and correct physico-chemical dependencies are needed to get correct SOA and its climate effects in future scenarios. Indeed, recent studies have been using the VBS or even older two-product schemes for future SOA prediction. The recent study by Cholakian et al. (2019) also pointed out that the two-product scheme is used for the simulation of SOA in most future scenarios, and they compared the two-product and the VBS schemes under the RCP8.5 scenario. Even more simplified SOA scheme was used in the recent climate study (a fixed yield (0.05) of the semi-volatile compound from isoprene oxidation, Sporre et al., 2019).

We changed the conclusion to put less emphasis on the need for explicit chemistry compared to the VBS scheme, as we agreed with the reviewer that the community has accepted the idea that the explicit chemistry is needed for accurate representation of SOA. Instead, we added more suggestions for future climate studies focusing on SOA changes based on sensitivity tests conducted in this study. We also added a more detailed discussion about future IEPOX-SOA changes based on the explicit scheme with quantitative numbers.

Line 684:

**"First, we evaluated the modeled IEPOX-SOA against aircraft campaigns over the SE US and the Amazon and global surface observations measured by AMS. The explicit treatment of IEPOX chemistry better captured both absolute concentrations and spatial/temporal variabilities than the VBS scheme. However, there is still room for improvement, as the model underestimated the observed IEPOX-SOA over the SE US while overestimated it over the Amazon. We further conducted sensitivity model runs to investigate the effects of recent findings from laboratory studies on modeled IEPOX-SOA concentrations. We concluded that a more detailed representation of IEPOX-SOA molecular tracers is needed for future models, because the chemical composition of IEPOX-SOA determines the volatility, aerosol pH, coating, and viscosity, and in turn, affects the formation of IEPOX-SOA."**

Line 700:

695 **"Our base model with the explicit chemistry predicted IEPOX-SOA increases in the 2090s - 45%, 97%, 173%, and 278%, under SSP1-2.6, SSP2-4.5, SSP3-7.0, and SSP5-8.5, respectively. For the SSP1-2.6, the predicted increase was lower than the isoprene emission (66%) and the VBS SOA (68%) increases. On the other hand, the base model simulated higher IEPOX-SOA increase than isoprene emission (187%) and the VBS SOA**
700 **changes (218%) under the SSP5-8.5. This is because** the explicit chemistry simulated 50% differences across different SSP scenarios, but the VBS scheme simulated nearly constant values in terms of the global SOA burden to isoprene emission ratio. **Therefore, IEPOX-SOA changes in previous studies that have used the VBS or two product schemes were likely overestimated under future scenarios with strong emission reductions (e.g., SSP1-2.6),**
705 **while underestimated under future scenarios with weak regulations (e.g., SSP3-7.0)."**

We also changed the portion of the title from "Future changes in isoprene-epoxydiol-derived secondary organic aerosol (IEPOX-SOA) under the shared socioeconomic pathways: **the importance of explicit chemistry**" to "Future changes in isoprene-epoxydiol-derived
710 secondary organic aerosol (IEPOX-SOA) under the shared socioeconomic pathways: **the importance of physico-chemical dependency**", in order to change the focus from "the explicit chemistry vs. the VBS scheme" to "physico-chemical dependency".

[revised manuscript text omitted]

**1 Isoprene emission comparison**

We compared the annual isoprene emissions for 2011–2013 simulated by CESM2.1.0/MEGANv2.1 and by OMI top-down estimate (Bauwens et al. 2016, available at: http://emissions.aeronomie.be/index.php/omi-based, last access: 1 March 2020). As shown in Figure S1c, CESM2.1.0/MEGANv2.1 overestimates isoprene emissions over the Tropics and underestimates them at high latitudes in the Northern Hemisphere. In terms of magnitude, isoprene emissions over the Tropics are important. We scaled down Tropical isoprene emissions by reducing emission factors of Tropical plant functional types (PFT). Two PFTs were used in the Community Land Model version 5 (CLM5): "broadleaf evergreen tropical tree" and "broadleaf deciduous tropical tree". These two PFTs contribute ~80% of total global isoprene emissions (Guenther et al. 2012). There were still regional discrepancies between CESM2.1.0/MEGANv2.1 and top-down estimates (Figures S1d and S1e), but the global total emission amount became closer to the total emission value by the OMI top-down estimate. Global total annual isoprene emissions changed from 439 Tg yr$^{-1}$ to 260 Tg yr$^{-1}$, which was comparable to the top-down estimate (266 Tg yr$^{-1}$).

[Figure]

**Figure S1.** Emission trajectories for $SO_2$ and $NO_x$ for the four Tier 1 scenarios used in this study.

[Figure]

**Figure S2.** Annual isoprene emissions for 2011–2013 by (a) OMI top-down and (b) CESM2.1.0/MEGANv2.1. (c) Ratios of CESM2.1.0/MEGANv2.1 to OMI top-down isoprene emissions. (d) same as (c) but the isoprene emission factors of tropical trees in CESM2.1.0/MEGANv2.1 are reduced by 50%. (e) Zonal mean cross-section of annual isoprene emissions.

[Figure]

**Figure S3.**  Time series of Isoprene emissions (2011–2013) over (a) Southeast US (30–40°N, 100–80°W), (b) Amazon (10–0°S, 70–60°W), (c) Africa (5°S–5°N, 10–30°E), and (d) Borneo (5°S–5°N, 105–120°E).

[Figure]

**Figure S4.** Histograms for observed and modeled (a) isoprene, (b) ISOPOOH, (c) IEPOX, and (d) IEPOX-SOA concentrations during the SEAC4RS campaign. Observation and model results are shown in black and blue bars, respectively. We note that gas measurements can be negative when the real concentrations are zero or very low due to instrumental noise. On the other hand, IEPOX-SOA concentrations were calculated by the positive matrix factorization method and always positive.

[Figure]

**Figure S4S5.** Same as Fig. 1 (c,d) but used different $H^*$ values. $8.5 \times 10^7$ , $8.5 \times 10^8$, $8.5 \times 10^9$ M atm$^{-1}$, are used for (a,d), (b,e), and (c,f), respectively. SOA yield from IEPOX reactive uptake was assumed to be 0.2. IEPOX comparisons are shown in top panels (a,b,c) and IEPOX-SOA in bottom panels (d,e,f).

[Figure]

**Figure S6.** Same as Fig. 2 (c) and (d) but used $H^*$ of $8.5 \times 10^7$ M atm$^{-1}$ and the yield of 0.2. The model with the half isoprene emission case (red) is only shown.

[Figure]

65  **Figure S7.** Same as Fig. S6 but assumed 90% of inorganic sulfates are converted to organic sulfates.

[Figure]

**Figure S8.** Annual mean aerosol pH at the surface for the year 2010 as simulated by CESM2 model. (a) Base case used in the paper (Base), (b) assuming 90% of inorganic sulfates are converted to organosulfates (Inor90), (c) difference between (a) and (b).

[Figure]

**Figure S9.** Histograms for observed and modeled (a,b) isoprene and (c,d) IEPOX-SOA concentrations during the GoAmazon campaign. Observations are shown in black bars, and CESM results with different isoprene emissions sensitivities are represented in blue (Base), red (Half), and green (Quarter) bars.

[Figure]

**Figure S10.** Same as Fig. 3 but for (a) the sensitivity model run with 90% of inorganic sulfate conversion to organic sulfate over the Amazon (black line), (b) the sensitivity model run with heterogeneous OH oxidation of IEPOX-SOA using the rate constant of $4.0 \times 10^{-13}$ cm$^3$ molec.$^{-1}$ s$^{-1}$ (blue line), and (c) IEPOX-SOA simulated by the VBS scheme (red line). For the VBS results, SOA from the isoprene + OH pathway was assumed as IEPOX-SOA. We assumed all aged IEPOX-SOA was lost via the fragmentation process for the heterogeneous OH reaction sensitivity run.

[Figure]

**Figure S11.** Simulated annual mean IEPOX-SOA concentrations at the surface. (a) Fresh IEPOX-SOA (b) Aged IEPOX-SOA (after heterogeneous oxidation against OH). The ratios between fresh and aged IEPOX-SOA are presented in panel (c). Aged IEPOX-SOA was assumed to be not evaporated and only lost via wet and dry depositions in the model.

[Figure]

**Figure S7S12.** Global ratios of OH simulated under future SSP scenarios (2090s) to present conditions (2010s) at the surface (Explicit case).

[Figure]

**Figure S13.** Global ratios of NO$_x$ under future SSP scenarios (2090s) to present conditions (2010s) at the surface (Explicit case).

[Figure]

95 **Figure S9S14.** Same as Fig. 6 but for absolute values of mass fluxes (in Tg yr⁻¹).

[Figure]

**Figure** **S15.** Global ratios of sulfate aerosols under future SSP scenarios (2090s) to present conditions (2010s) at the surface (Explicit case).

[Figure]

100 **Figure S11S16.** The global aerosol pH (accumulation mode) differences between future SSP scenarios (2090s) and present conditions (2010s) at the surface (Explicit case).

[Figure]

**Figure  S17.** Simulated IEPOX-SOA concentration changes at the surface by including sea salt in aerosol thermodynamic calculation (EXP_SS_2090 - EXP_2090).

[Figure]

**Figure S18.** The multi-year mean global aerosol pH (accumulation mode) of EXP_SS_2090 simulations at the surface (left column) and pH differences between EXP_SS_2090 and EXP_2090 simulations.

[Figure]

110 **Figure S19.** Global ratio maps of surface mean IEPOX-SOA concentrations between EXP_2090_CO2 (with $CO_2$ inhibition) and EXP_2090 (without $CO_2$ inhibition) for different SSP scenarios.

[Figure]

**Figure S20.** Global maps of activity factor changes (ratio of SSP1-2.6 to present) used in isoprene emission calculations by MEGANv2.1. (a) total activity factor ($\gamma$), (b) leaf area index (LAI), (c) emission response to light ($\gamma_P$), (d) temperature ($\gamma_T$), (e) leaf age ($\gamma_A$), and (f) $CO_2$ inhibition. See Eq. (2) in Guenther et al. (2012) for details. Soil moisture factor is not included, as CESM2.1.0 applies a unity value for soil moisture factor.

[Figure]

**Figure S14S21.** Same as Fig. S13S20 but for the SSP5-8.5 scenario.

[Figure]

**Figure S22.** Global ratio maps of OH concentrations between simulations with (EXP) and without $CO_2$ (EXP_CO2) inhibition effects.

**Table S1.** A brief summary of SSP scenarios used in this study. Values are for the year 2100. For more information, readers are referred to previous studies (O'Neill et al., 2016; Riahi et al., 2017; Kc and Lutz, 2017; Gidden et al., 2019; Feng et al., 2020).

| | | |
|---|---|---|
| SSP1-2.6 | Title[1] | Sustainability - Taking the Green Road (Low challenges to mitigation and adaptation) |
| | Description[1] | The world shifts gradually, but pervasively, toward a more sustainable path, emphasizing more inclusive development that respects perceived environmental boundaries. Management of the global commons slowly improves, educational and health investments accelerate the demographic transition, and the emphasis on economic growth shifts toward a broader emphasis on human well-being. Driven by an increasing commitment to achieving development goals, inequality is reduced both across and within countries. Consumption is oriented toward low material growth and lower resource and energy intensity. |
| | Forcing category[2] | Low |
| | Target forcing level[2] ($W\ m^{-2}$) | 2.6 |
| | Population[3] (millions) | 6,881 |
| | Land use change regulation[1] | strong |
| | Sulfur emissions[4] ($Mt\ SO_2\ yr^{-1}$) | 8.1 |
| | $NO_x$ emissions[4] ($Mt\ NO_2\ yr^{-1}$) | 41.2 |
| | VOC emissions[4] ($Mt\ VOC\ yr^{-1}$) | 62.3 |
| | OC emissions[4] ($Mt\ OC\ yr^{-1}$) | 13.1 |
| | $CO_2$ emissions[4] ($Mt\ CO_2\ yr^{-1}$) | -8,618 |
| SSP2-4.5 | Title[1] | Middle of the Road (Medium challenges to mitigation and adaptation) |

| | Description[1] | The world follows a path in which social, economic, and technological trends do not shift markedly from historical patterns. Development and income growth proceeds unevenly, with some countries making relatively good progress while others fall short of expectations. Global and national institutions work toward but make slow progress in achieving sustainable development goals. Environmental systems experience degradation, although there are some improvements and overall the intensity of resource and energy use declines. Global population growth is moderate and levels off in the second half of the century. Income inequality persists or improves only slowly and challenges to reducing vulnerability to societal and environmental changes remain. |
|---|---|---|
| | Forcing category[2] | Medium |
| | Target forcing level[2] (W m$^{-2}$) | 4.5 |
| | Population[3] (millions) | 9,000 |
| | Land use change regulation[1] | medium |
| | Sulfur emissions[4] (Mt SO$_2$ yr$^{-1}$) | 30.8 |
| | NO$_x$ emissions[4] (Mt NO$_2$ yr$^{-1}$) | 77.7 |
| | VOC emissions[4] (Mt VOC yr$^{-1}$) | 120.7 |
| | OC emissions[4] (Mt OC yr$^{-1}$) | 14.5 |
| | CO$_2$ emissions[4] (Mt CO$_2$ yr$^{-1}$) | 9,683 |
| SSP3-7.0 | Title[1] | Regional Rivalry – A Rocky Road (High challenges to mitigation and adaptation) |

| | | |
|---|---|---|
| | Description[1] | A resurgent nationalism, concerns about competitiveness and security, and regional conflicts push countries to increasingly focus on domestic or, at most, regional issues. Policies shift over time to become increasingly oriented toward national and regional security issues. Countries focus on achieving energy and food security goals within their own regions at the expense of broader-based development. Investments in education and technological development decline. Economic development is slow, consumption is material-intensive, and inequalities persist or worsen over time. Population growth is low in industrialized and high in developing countries. A low international priority for addressing environmental concerns leads to strong environmental degradation in some regions. |
| | Forcing category[2] | High |
| | Target forcing level[2] (W m$^{-2}$) | 7.0 |
| | Population[3] (millions) | 12,627 |
| | Land use change regulation[1] | weak |
| | Sulfur emissions[4] (Mt $SO_2$ yr$^{-1}$) | 78.1 |
| | NO$_x$ emissions[4] (Mt $NO_2$ yr$^{-1}$) | 144.4 |
| | VOC emissions[4] (Mt VOC yr$^{-1}$) | 227.9 |
| | OC emissions[4] (Mt OC yr$^{-1}$) | 33.7 |
| | $CO_2$ emissions[4] (Mt $CO_2$ yr$^{-1}$) | 82,726 |
| SSP5-8.5 | Title[1] | Fossil-fueled Development – Taking the Highway (High challenges to mitigation, low challenges to adaptation) |

| | | |
|---|---|---|
| Description[1] | This world places increasing faith in competitive markets, innovation and participatory societies to produce rapid technological progress and development of human capital as the path to sustainable development. Global markets are increasingly integrated. There are also strong investments in health, education, and institutions to enhance human and social capital. At the same time, the push for economic and social development is coupled with the exploitation of abundant fossil fuel resources and the adoption of resource and energy intensive lifestyles around the world. All these factors lead to rapid growth of the global economy, while global population peaks and declines in the 21st century. Local environmental problems like air pollution are successfully managed. There is faith in the ability to effectively manage social and ecological systems, including by geo-engineering if necessary. | |
| Forcing category[2] | High | |
| Target forcing level[2] (W m$^{-2}$) | 8.5 | |
| Population[3] (millions) | 7,363 | |
| Land use change regulation[1] | medium | |
| Sulfur emissions[4] (Mt $SO_2$ yr$^{-1}$) | 29.5 | |
| $NO_x$ emissions[4] (Mt $NO_2$ yr$^{-1}$) | 98.7 | |
| VOC emissions[4] (Mt VOC yr$^{-1}$) | 163.3 | |
| OC emissions[4] (Mt OC yr$^{-1}$) | 17.6 | |
| $CO_2$ emissions[4] (Mt $CO_2$ yr$^{-1}$) | 126,287 | |

1) Riahi et al. (2017); 2) O'Neill et al. (2016); 3) KC and Lutz (2017); 4) Gidden et al. (2019)

Table S2. Datasets used in Sect. 3.3 and Fig. 4[a]. Ranges or average plus standard deviation of $C_5H_6O$ (high resolution) and $f_{82}$ (unit mass resolution) in different studies are also included.

| Name of datasets | Time Period | Site locations and descriptions | Campaign name | Ranges or average±std.dev. $f_{C5H6O}$ (‰) | Ranges or average±std.dev. $f_{82}$ (‰) | OA Conc. (ug/m3) | IEPOX-SOA Conc. (ug/m3) | IEPOX-SOA/OA (%) | Latitude | longitude | Ref. | X axis label in Fig. 4 |
|---|---|---|---|---|---|---|---|---|---|---|---|---|
| **Studies strongly-influenced by isoprene emissions under lower NO** | | | | | | | | | | | | |
| SE US forest - CTR site, 2013 SOAS | Jun-Jul, 2013 | Centreville, AL | SOAS | 6.2±2.4 | 7.6±2.2 | 3.8 | 0.64 | 17 | 32.95 | -87.13 | -1 | Centreville 2013 Summer |
| SE US forest - Look Rock site, 2013 SOAS | Jun-Jul, 2013 | Look Rook | SOAS | N/A | N/A | 4.87 | 1.6 | 33 | 35.61 | -83.55 | -2 | N/A[c] |
| SE US forest - Look Rock site, 2013 Spring | Mar-May, 2013 | Look Rock | N/A | N/A | N/A | 3.23 | 1.32 | 41 | 35.61 | -83.55 | -3 | Look Rock, 2013 Spring |
| SE US forest - Look Rock site, 2013 Summer | Jun-Sep, 2013 | Look Rock | N/A | N/A | N/A | 5.32 | 2.13 | 40 | 35.61 | -83.55 | -3 | Look Rock, 2013 Summer |
| SE US forest - Look Rock site, 2013 Fall | Oct-Dec, 2013 | Look Rock | N/A | N/A | N/A | 2.83 | 0.76 | 27 | 35.61 | -83.55 | -3 | Look Rock, 2013 Fall |
| Atlanta JST site, 2012 Spring | Mar-Jun, 2012 | Urban JST site, Atlanta, Georgia, US | N/A | N/A | N/A | 4.7 | 1.74 | 37 | 33.78 | -84.42 | -3 | Atlanta (JST), 2012 Spring |
| Atlanta JST site, 2013 Summer | Jul-Sep, 2013 | Urban JST site, Atlanta, Georgia, US | N/A | N/A | N/A | 6.15 | 2.34 | 38 | 33.78 | -84.42 | -3 | Atlanta (JST), 2013 Summer |
| Atlanta JST site, 2014 Spring | May-Jun, 2014 | Urban JST site, Atlanta, Georgia, US | N/A | N/A | N/A | 9.61 | 2.4 | 25 | 33.78 | -84.42 | -4 | Atlanta (JST), 2014 Spring |
| Atlanta JST site, 2014 Summer | Jul-Sep, 2014 | Urban JST site, Atlanta, Georgia, US | N/A | N/A | N/A | 11.36 | 3.29 | 29 | 33.78 | -84.42 | -4 | Atlanta (JST), 2014 Summer |
| Pristine Amazon forest 2008, Brazil | Feb-Mar, 2008 | Pristine rain forest site, TT34 | AMAZE-08 | 5.0±2.3 | 7.9±1.7 | 0.76 | 0.26 | 34 | -2.59 | -60.2 | -5 | Amazon, 2008 Summer |

| | | | | | | | | | | | | |
|---|---|---|---|---|---|---|---|---|---|---|---|---|
| Amazon forest downwind Manaus, Brazil | Feb-Mar, 2014 | T3 site, near Manacapuru | GoAmazon2014/5 | 6.9±1.6 | 7.1±1.0 | 1.3 | 0.286 | 22 | -3.21 | -60.59 | -6 | Amazon, 2014 Summer |
| Pristine Amazon forest 2014, Brazil | Aug-Dec, 2014 | T0 site, ~150 km northeast of Manaus | GoAmazon2014/5 | N/A | 5.6±1.7 | N/A | N/A | N/A | -3.21 | -60.59 | -7 | N/A |
| SEAC4RS | Aug-Sep, 2013 | Aircraft measurement | SEAC4RS | 4.3±1.6 | N/A | N/A | N/A | 32 | Flight track | Flight track | -8 | N/A |
| Borneo forest, Malaysia | Jun-Jul, 2008 | Rain forest GAW station, Sabah, Malaysia | OP3 | 10±0.3 | 12.4±0.4 | 0.75 | 0.18 | 24 | 4.981 | 117.844 | -9 | Borneo, 2008 Summer |
| Atlanta JST site, 2011 Summer | Aug-Sep, 2011 | Urban JST site, Atlanta, Georgia, US | N/A | N/A | 3.7±1.9 | 11.6 | 3.8 | 33 | 33.78 | -84.42 | -10 | Atlanta (JST), 2011 Summer |
| Atlanta JST site, 2012 May | May, 2012 | Urban JST site, Atlanta, Georgia, US | N/A | 3.3±0.9 | N/A | 9.1 | 1.91 | 21 | 33.78 | -84.42 | -11 | N/A[d] |
| Atlanta GT site, 2012 Summer | Aug, 2012 | Urban Georgia Tech site, Georgia, US | N/A | 5.4±1.9 | N/A | 9.6 | 3 | 31 | 33.78 | -84.396 | -11 | Atlanta (GT), 2012 Summer |
| Yorkville, 2012 Summer | July, 2012 | Rural sites, 80km northwest of JST site, Georgia, US | N/A | 7.7±2.2 | N/A | 11.2 | 4 | 36 | 33.9285 | -85.045 | -11 | Yorkville, 2012 Summer |
| Harrow, Canada | Jun-Jul, 2007 | Harrow site, rural sites surrounded by farmland, Canada | BAQSMET | N/A | N/A | N/A | N/A | 17 | 42.03 | -82.9 | -12 | N/A |
| Bear Creek, Canada | Jun-Jul, 2007 | Bear Creek site, wetlands area surrounded by farmland, Canada | BAQSMET | N/A | N/A | N/A | N/A | 6 | 42.51 | -82.34 | -12 | N/A |
| **Studies strongly-influenced by monoterpene emissions** | | | | | | | | | | | | |
| Rocky mountain pine forest, CO, USA | Jul-Aug, 2011 | Manitou Experimental Forest Observatory, CO, | BEACHON-RoMBAS | 3.7±0.5 | 5.1±0.5 | N/A | N/A | N/A | 39.1 | -105.1 | -13 | N/A |
| European Boreal forest, Finland | 2008-2009 | Hyytiala site in Pine forest, Finland | EUCAARI campaign | 2.5±0.1 [b] | 4.8±0.1 [b] | N/A | N/A | N/A | 61.85 | 24.28 | -9 | N/A |

| | | | | Studies mixed-influenced by isoprene and monoterpene emissions | | | | | | | | |
|---|---|---|---|---|---|---|---|---|---|---|---|---|
| North American temperate, US | Aug-Sep, 2007 | Blodgett Forest Ameriflux Site, CA, US | BEARPEX | 4.0±<0.1 [b] | 4.0±<0.1 [b] | N/A | N/A | N/A | N/A | N/A | -9 | N/A |
| **Studies strongly-influenced by urban emissions** | | | | | | | | | | | | |
| Los Angeles area , CA, USA | May-Jun, 2010 | Pasadena, US | CalNex | 1.6±0.2 | 3.6±0.5 | 7 | <DL | < PMF limit | 34.14 | -118.12 | -14 | Pasadena 2010 Spring |
| Beijing, China | Nov-Dec, 2010 | Peking University, in NW of Beijing city, China | N/A | 1.5±0.3 | 4.6±0.7 | 34.5 | <DL | < PMF limit | 39.99 | 116.31 | -15 | Beijing 2010 Winter |
| Changdao island, Downwind of China | Mar-Apr, 2011 | Changdao island, China | CAPTAIN | 1.6±0.2 | 3.8±0.5 | 13.4 | <DL | < PMF limit | 37.99 | 120.7 | -16 | Changdao 2011 Spring |
| Barcelona area, Spain | Feb-Mar, 2009 | Montseny, Spain | DAURE | 1.6±0.2 | 4.8±0.9 | N/A | <DL | < PMF limit | 41.38 | 2.1 | -17 | Montseny 2009 Spring |

a- HR-ToF-AMS was used for all the campaigns except the Atlanta, US, Look Rock, US, and Pristine Amazon forest 2014, Brazil using ACSM.

b- Standard error

c- included in Look Rock 2013 Summer

d- included in Atlanta (JST) 2012 Spring

(1)(Hu et al., 2015b); (2)(Budisulistiorini et al., 2015); (3)(Budisulistiorini et al., 2016); (4)(Rattanavaraha et al., 2017); (5)(Chen et al., 2014); (6)(de Sá et al., 2017); (7)(Carbone et al., 2015);

(8)(Liao et al., 2014) ; (9)(Robinson et al., 2011); (10)(Budisulistiorini et al., 2013); (11)(Xu et al., 2014; Xu et al., 2015); (12)(Slowik et al., 2011); (13)(Ortega et al., 2014); (14)(Hayes et al., 2013);

(15)(Hu et al., 2015a); (16)(Hu et al., 2013); (17)(Minguillón et al., 2011)